

# Assessing the Impact of Clouds on UV-visible Total Column Ozone Measurements in the High Arctic

Xiaoyi Zhao[1,2], Kristof Bognar[1], Vitali Fioletov[2], Andrea Pazmino[3], Florence Goutail[3], Luis Millán[4], Gloria Manney[5,6], Cristen Adams[7], Kimberly Strong[1]

[1]Department of Physics, University of Toronto, Toronto, Ontario, M5S 1A7, Canada.
[2]Measurement and Analysis Research Section, Environment and Climate Change Canada, Toronto, M3H 5T4, Ontario, Canada.
[3]Versailles St-Quentin, CNRS/INSU, LATMOS-IPSL, 78280 Guyancourt, France.
[4]Jet Propulsion Laboratory, California Institute of Technology, Pasadena, California, USA.
[5]NorthWest Research Associates, Socorro, New Mexico, USA.
[6]Department of Physics, New Mexico Institute of Mining and Technology, Socorro, New Mexico, USA.
[7]Environmental Monitoring and Science Division, Government of Alberta, Edmonton, Alberta, Canada.

*Correspondence to*: Xiaoyi Zhao (xiaoyi.zhao@canada.ca or xizhao@atmosp.physics.utoronto.ca)

**Abstract.** Zenith-Sky scattered light Differential Optical Absorption Spectroscopy (ZS-DOAS) has been used widely to retrieve total column ozone (TCO). ZS-DOAS measurements have the advantage of being less sensitive to clouds than direct-sun measurements. However, the presence of clouds still affects the quality of ZS-DOAS TCO. Clouds are thought to be the largest contributor to random uncertainty in ZS-DOAS TCO, but their impact on data quality still needs to be quantified. This study has two goals: (1) to investigate whether clouds have a significant impact on ZS-DOAS TCO, and (2) to develop a cloud-screening algorithm to improve ZS-DOAS measurements in the Arctic under cloudy conditions. To quantify the impact of weather, eight years of measured and modelled TCO have been used, along with information about weather conditions at Eureka, Canada (80.05°N, 86.41°W). Relative to direct-sun TCO measurements by Brewer spectrophotometers and modelled TCO, a positive bias is found in ZS-DOAS TCO measured in cloudy weather, and a negative bias is found for clear conditions, with differences of up to 5% between clear and cloudy conditions. A cloud-screening algorithm is developed for high-latitudes using the colour index calculated from ZS-DOAS spectra. The quality of ZS-DOAS TCO datasets is assessed using a statistical uncertainty estimation model, which suggests a 3-4% random uncertainty. The new cloud-screening algorithm reduces the random uncertainty by 0.6%. If all measurements collected during cloudy conditions, as identified using the weather station observations, are removed, the random uncertainty is reduced by 1.3%. This work demonstrates that clouds are a significant contributor to uncertainty in ZS-DOAS TCO and proposes a method that can be used to screen clouds in high-latitude spectra.



## 1 Introduction

Ozone is one of the most widely monitored trace gases in the atmosphere. It can be measured via its strong absorption bands in the ultraviolet (UV), visible (Vis) and infrared (IR) portions of the spectrum. Remote sensing measurements of total column ozone (TCO) started in the 1920s with the Dobson instrument (Dobson, 1968), which measures the UV spectrum

(the so-called Huggins bands). During the International Geophysical Year, 1957-58, the worldwide Dobson ozone-monitoring network was established. Stratospheric ozone has been a focus of scientific study since the 1970s and became a matter of intense interest with the discovery and subsequent studies of the Antarctic ozone hole (Farman et al., 1985; Solomon et al., 1986; Stolarski et al., 1986) and depletion on the global scale (Stolarski et al., 1991; Ramaswamy et al., 1992).

To improve the accuracy of, and to automate, TCO measurements, the Brewer spectrophotometer was developed in the early 1980s (Kerr et al., 1981, 1988). In 1988, the Brewer was designated (in addition to the Dobson) as the World Meteorological Organization (WMO) Global Atmosphere Watch (GAW) standard for TCO measurement. By 2017, there were more than 230 Brewer instruments installed around the world. Brewer instruments can provide TCO values via four types of observations: direct-sun, direct-moon, zenith-sky, and spectral UV irradiance (De Backer and De Muer, 1991; Fioletov et al.,

1997, 1999; Labow et al., 2013). The most accurate ozone data products from Brewer instruments are their direct sun (DS) measurements, which have a typical accuracy of 1% (Fioletov et al., 2005). One limitation of Dobson/Brewer UV instruments is the so-called stray light effect (Kerr et al., 1981; Van Roozendael et al., 1998; Fioletov et al., 2000), which prevents the use of Dobson/Brewer instruments to retrieve TCO at large solar zenith angles (SZAs, above 80°).

Since the 1990s, a zenith-sky UV-visible ozone monitoring group has been operating within the Network for the Detection

of Atmospheric Composition Change (NDACC) (Sarkissian et al., 1995; Vaughan et al., 1997; Van Roozendael et al., 1998; Van Roozendael and Hendrick, 2009; Hendrick et al., 2011). Unlike Dobson/Brewer instruments, NDACC UV-visible instruments use the zenith-sky visible spectrum (Chappuis bands) to retrieve TCO. The use of visible spectroscopy makes it possible to measure TCO at higher SZAs (up to 91°), which allows for the collection of data at high latitudes during polar sunrise and sunset. The NDACC UV-visible network consists of more than 35 instruments that have provided more than two

decades of measurements of total column amounts of ozone, $NO_2$, BrO, and OClO retrieved using the zenith-sky scattered sunlight differential optical absorption spectroscopy (ZS-DOAS) technique (Vaughan et al., 1997; Van Roozendael et al., 1998; Hendrick et al., 2011). A UV-visible ZS-DOAS instrument, the University of Toronto Ground-based Spectrometer (UT-GBS) has been deployed in Eureka, Nunavut, Canada (80.05°N, 86.41°W) during springtime from 1999 to 2009 and year-round since 2010, and it is part of the NDACC UV-visible network. In addition, an NDACC-certified Système

D'Analyse par Observations Zénithales (SAOZ) instrument has been deployed at the same site since 2005. Both UT-GBS and SAOZ data analyses follow the NDACC retrieval protocols (Van Roozendael and Hendrick, 2009) and use the NDACC ozone air mass factor (AMF) look-up table (LUT) in the TCO retrieval.



Many studies have compared WMO/GAW Dobson/Brewer TCO (hereafter referred to as DB TCO) with NDACC UV-visible zenith-sky TCO (referred to as ZS TCO) (Kyrö, 1993; Roscoe et al., 1994, 2001; Høiskar et al., 1997; Vaughan et al., 1997; Van Roozendael et al., 1998; Fraser et al., 2007; Hendrick et al., 2011). In general, it has been found that ZS TCO retrievals have advantages such as weak temperature dependence of ozone cross sections (in the visible band), the ability to

measure at large SZA (e.g., during polar sunrise and sunset), and limited sensitivity to clouds compared to DB TCO (Daumont et al., 1992; Scarnato et al., 2009; Van Roozendael and Hendrick, 2009; Hendrick et al., 2011). However, ZS TCO also has characteristics such as low temporal coverage (twice per day), low total accuracy (6%, compared to 1% for DB TCO), and dependency on the AMF calculated using a radiative transfer model (RTM) (Wardle, 1997; Van Roozendael et al., 1998; Van Roozendael and Hendrick, 2009; Hendrick et al., 2011; Zhao et al., 2016b). Van Roozendael et al. (1998)

reported that the sensitivity of ZS ozone AMFs to multiple scattering in tropospheric clouds could lead to occasional positive bias in ZS TCO retrieved from SAOZ instruments. Hendrick et al. (2011) concluded that the main sources of uncertainties in the ZS ozone AMF calculation are: (1) inaccurate ozone profiles and surface albedo, (2) the choice of aerosol extinction profile and RTM, and (3) the presence of clouds.

However, clouds are not accounted for in the NDACC ozone AMF calculations (Hendrick et al., 2011). This is because the

twilight zenith-sky measurements are strongly weighted by the contribution of the stratospheric ozone and therefore show limited sensitivity to the uncertainties in parameters affecting tropospheric ozone (e.g., Mie scattering in a cloud layer) (Hendrick et al., 2011). Hendrick et al. (2011) reported that cloudy AMFs are systematically larger than non-cloudy AMFs by about 5-8% at 86° SZA and 2% at 91° SZA. This leads to a random uncertainty of 3.3% for TCO calculated using the NDACC ozone AMF LUT between 86-91° SZA. In fact, clouds are the largest source of random uncertainty in ZS TCO.

The second largest source, the climatological ozone profile, only accounts for 1%, and the third largest source, aerosols, only accounts for 0.6%. Based on the uncertainty budget (Table 4) in Hendrick et al. (2011), ZS TCO precision is 4.7%. Theoretically, it could be improved to 3.4% if the uncertainty due to cloud was removed. Sarkissian et al. (1997) found that low-altitude clouds have a very small effect on ozone AMFs, and there was no systematic deviation of the TCOs measured by SAOZ relative to ozonesondes when total cloud cover was observed. Pfeilsticker et al. (1998) categorized cloud effects

on the basis of three processes (geometry effect, multiple reflection effect, and photon diffusion) and quantified their magnitudes using RTM calculations. They reported that these processes may introduce significant errors in ZS TCO. Pfeilsticker et al. (1998) shows that the enhanced ozone absorption due to photon diffusion in the cloud may increase the ZS TCO by as much as 9%. It is clear that different types of clouds (different cloud optical depth, height, water or ice content, etc.) can have different impacts on ZS-DOAS TCO accuracy.

While ZS-DOAS measurements are affected by clouds, the Multi-Axis DOAS (MAX-DOAS) technique (Sanders et al., 1993; Platt, 1994; Hönninger et al., 2004; Platt and Stutz, 2008) is even more sensitive to clouds. Unlike ZS-DOAS, which measures at only 90° elevation viewing angle, MAX-DOAS measures over a range of elevation angles (typically 3-10 different angles, from 0° to 90°). At low elevation angles, sunlight arriving at the instrument has typically taken a long path through the troposphere and hence has greater sensitivity for tropospheric trace gases (Platt and Stutz, 2008). This enhanced





tropospheric sensitivity also creates an urgent need for a cloud and aerosol detection and classification algorithm for MAX-DOAS measurements (Gielen et al., 2014; Wagner et al., 2014, 2016; Wang et al., 2015). In general, these algorithms are based on the colour index (CI, the intensity ratio of two measured wavelengths) and $O_4$ absorption derived from ZS/MAX-DOAS measurements at mid-latitudes (more details are provided in Section 3.1). However, at this time, there is no cloud-

screen (detection) algorithm developed specifically for ZS-DOAS measurements at high latitudes, where the limited SZA range makes it challenging to apply any of the previously developed algorithms. For example, the algorithm developed by Wagner et al. (2016) needs measurements when SZA < 60°, whereas these small SZA measurements only account for about 7% of UT-GBS year-round measurements at Eureka, which is located at 80° N, where the lowest SZA is about 56°.

The objective of this work is to develop a cloud detection algorithm for high-latitude measurements using data collected by

ZS-DOAS instruments deployed at Eureka to improve TCO data quality. This paper is organized as follows. Section 2 describes the measured and modelled TCO data used in this study, with additional information about Eureka weather records. In Section 3, by adapting and improving some cloud-screen algorithms from MAX-DOAS instruments, a new algorithm for high-latitude ZS-DOAS measurements is introduced. This algorithm is applied to UT-GBS and SAOZ TCO retrievals, to help identify the weather conditions during the measurements and to improve measurement accuracy. In

Section 4, both the standard and cloud-screened ZS-DOAS TCO data are compared to Brewer direct-sun and modelled TCO data. Random uncertainties are estimated for all ZS-DOAS TCO datasets using a statistical uncertainty estimation model. A discussion of scientific significance and conclusions is given in Section 5. In short, by generating long-term ozone time series that are unbiased by meteorological conditions, this work will help the validation of satellite algorithms for cloudy scenes (Fioletov et al., 2011). In the future, this high-quality ground-based TCO dataset will be used for satellite validation

in the high Arctic.

## 2 Datasets and models

### 2.1 UT-GBS

The UT-GBS is a Triax-180 grating spectrometer, built by Jobin-Yvon/Horiba. The Triax-180 is a crossed Czerny-Turner triple grating imaging spectrometer. Light is directed by a collimating mirror to a grating and is then focused by a focusing

mirror onto a charge-coupled device (CCD) detector. This instrument was assembled in 1998 and has been involved in numerous field campaigns summarised in Zhao (2017). These include the MANTRA 1998 balloon campaigns in Vanscoy, Saskatchewan (Bassford et al., 2001, 2005) and the 2009 CINDI campaign (Roscoe et al., 2010) at Cabauw, the Netherlands. When it is not travelling, the UT-GBS takes measurements in the University of Toronto Atmospheric Observatory or stays at the Polar Environment Atmospheric Research Laboratory (PEARL) at Eureka (Fogal et al., 2013; Zhao et al., 2016a).

Over the last 18 years, several components of the instrument have been changed. The field-of-view (FOV) of the instrument was changed from 2° to 0.2° in 2012 after an upgrade of the input optics (Zhao, 2017). The instrument was upgraded to a



ZS/MAX-DOAS system by coupling with a solar-tracker system in 2015 (Franklin, 2015; Zhao, 2017). In 2011, a reprocessed TCO dataset (1999-2011) with the NDACC AMF LUT version 1.0, was used for satellite validation (Adams et al., 2012). In the current work, the latest NDACC AMF LUT (version 2.0) is used in the TCO retrieval.

In this work, UT-GBS measurements made at the PEARL Ridge Lab from 2010 to 2017 are used. For this period, the UT-GBS was operated with a 600 groove per millimeter grating, and recorded spectra between 350 and 560 nm with resolution of 0.4-2 nm (Adams, 2012; Zhao, 2017). The UV-visible spectra were processed using the QDOAS software (Danckaert et al., 2015) using daily reference spectra. Due to the decreased resolution at the edge of CCD, the ozone differential slant column densities (dSCDs) were retrieved in the 450-545 nm window, instead of the NDACC recommended 450-550 nm window. Following the NDACC recommendations (Van Roozendael and Hendrick, 2009), cross sections of ozone (Burrows et al., 1999), $NO_2$ (Vandaele et al., 1998), $H_2O$ (Rothman et al., 2005), $O_4$ (Greenblatt et al., 1990), and Ring (Chance and Spurr, 1997) were all fitted, and a third-order polynomial was included in the DOAS analysis. The accuracy of GBS TCO data in the high Arctic (2003-2011) is 6.2 % (Adams, 2012; Adams et al., 2012).

A new cloud-screening TCO retrieval package was developed for UT-GBS ZS-DOAS measurements, to convert ozone dSCDs to vertical column densities (VCDs). Two versions of UT-GBS data are discussed in this work: (1) NDACC standard ZS-DOAS TCO data (referred to as GBS data), and (2) cloud-screened ZS-DOAS TCO data ($GBS_{CS}$ data). Details of the data processing are provided in Section 3.

**2.2 SAOZ**

The first SAOZ instrument was constructed in the late 1980s and designed as a ZS-DOAS instrument (Pommereau and Goutail, 1988). SAOZ records spectra between 270 and 620 nm, with a resolution of 1 nm. Two SAOZ instruments have performed measurements at Eureka since 2005. SAOZ no. 15 was deployed at the PEARL Ridge Lab from 2005 to 2009 for springtime measurements, and SAOZ no. 7 has been deployed since 2010 for year-round sunlit measurements. SAOZ and UT-GBS TCO data have been compared during several mid-latitude and Arctic campaigns (Fraser et al., 2007, 2008, 2009; Roscoe et al., 2010; Adams et al., 2012).

In this work, the UT-GBS cloud-screening TCO retrieval algorithm was used to retrieve SAOZ TCO. Two versions of SAOZ data were generated: (1) NDACC standard ZS-DOAS TCO data (referred to as SAOZ), and (2) cloud-screened data ($SAOZ_{CS}$). The SAOZ and $SAOZ_{CS}$ data all used the same ozone dSCDs provided by LATMOS, in the NDACC recommended 450-550 nm window.

The accuracy of SAOZ TCO was estimated to be 0-9% (Roscoe et al., 1994, 2001; Sarkissian et al., 1997) before the standardized NDACC ozone retrieval protocol was implemented. The accuracy of NDACC/SAOZ TCO data at mid-latitudes is reported to be 5.9 % (Hendrick et al., 2011). Details of the SAOZ data processing can be found in Section 3.



## 2.3 Brewer

The Brewer instruments use a holographic grating in combination with a slit mask to select six channels in the UV (303.2, 306.3, 310.1, 313.5, 316.8, and 320 nm) to be detected by a photomultiplier (Kerr, 2002). The first and second wavelengths are used for internal calibration and measuring $SO_2$, respectively. The four longer wavelengths are used for the ozone

retrieval. The TCO is calculated by analyzing the relative intensities at these different wavelengths using the Bass and Paur (1985) ozone cross section.

Four Brewer instruments (no. 21, 69, 111, and 192) have been deployed at Eureka since 1992 by Environment and Climate Change Canada (ECCC). Brewer no. 69, an MKV monochromator, took measurements from 1992-2017 (the longest Brewer TCO record at Eureka). During the time of this study, Brewer no. 69 was located on the roof of the Eureka weather station

main building, which is 15 km away from the PEARL Ridge Lab. In this work, Brewer no. 69 direct-sun spectra were analysed using the standard Brewer algorithm (Kerr et al., 1981), with small changes to the analysis parameters due to the high latitude of the measurements (Adams et al., 2012). This Brewer TCO dataset is referred to as Brewer. The random uncertainty of Brewer data is typically less than 1% (Fioletov et al., 2005), and for high-quality data (e.g., SZA < 71°) it is less than 0.6% (Zhao et al., 2016b).

## 2.4 MERRA-2

The second Modern-Era Retrospective analysis for Research and Applications (MERRA-2) is an atmospheric reanalysis from NASA's Global Modeling and Assimilation Office (GMAO) that provides high-resolution globally gridded meteorological fields using the Goddard Earth Observing System-Version 5 data assimilation system (Bosilovich et al., 2015; Fujiwara et al., 2017; Gelaro et al., 2017). MERRA-2 has a horizontal resolution of 0.625° × 0.5° (longitude ×

latitude). In this work, vertical profiles of MERRA-2 ozone (Wargan et al., 2017), temperature, pressure, and scaled potential vorticity (sPV) over Eureka were computed using the Jet and Tropopause Products for Analysis and Characterization (JETPAC) package described by Manney et al (2011, 2017). The sPV is potential vorticity scaled in "vorticity units" to give a similar range of values at each level, which can be used to identify the location of the polar vortex (Dunkerton and Delisi, 1986; Manney et al., 1994; Adams et al., 2013; Zhao et al., 2017). The profile data are on 72 model levels with 3-hour

temporal resolution and approximately 1-km vertical spacing near the tropopause.

The MERRA-2 TCO at Eureka has been used in a previous study by Zhao et al. (2017) to supplement Brewer TCO. In that study, the MERRA-2 TCO (2005-2015) for Eureka has a strong correlation (R = 0.99) and a small positive bias (1.6 %) compared to Brewer TCO. For the current work, the use of MERRA-2 TCO provides important information because: (1) MERRA-2 has 3-hour temporal resolution, and therefore MERRA-2 TCO can match ZS TCO (observations made when the

SZA is in range 86°-91°) more closely in time than DB TCO (Brewer observes TCO when SZA < 82°), and (2) MERRA-2 has continuous TCO data, which is not limited by sunlight or weather (cloud) conditions (whereas, Brewer data start in late March, and are limited to cloud-free conditions). Thus, MERRA-2 TCO can be used to assess the cloud impact on ZS TCO,





and to estimate the resulting statistical uncertainty (which needs large sample size; more details are provided in Section 4.2). In this study, MERRA-2 TCO data from 2010 to 2017 have been used.

The MERRA-2 data were also used to identify the location of polar vortex, as it can have a non-negligible impact on the TCO measurements. For example, when the polar vortex is present, it is possible that the zenith-sky observations sampled ozone-depleted air within the vortex, while the direct-sun observations measured ozone-rich air outside the vortex (e.g., Adams et al., 2012), or vice versa. Following Adams (2012), the MERRA-2 sPV profile was interpolated to the 490 K potential temperature level (lower stratosphere ozone maximum, for Eureka, about 17-21 km) and is referred to here as $sPV_{490K}$. The inner and outer vortex edges are identified at $sPV_{490K}$ values of $1.6 \times 10^{-4}$ $s^{-1}$ and $1.2 \times 10^{-4}$ $s^{-1}$ (Manney et al., 2007) respectively. For the eight-year period of this study (2010-2017), about 10% of ZS TCO measurements were made when the polar vortex was above Eureka. However, only 1% of the coincident ZS and DB TCO measurements were made when the vortex was above Eureka. Further details about the impact of the polar vortex are presented in Section 4.1.

## 2.5 Eureka Weather Station meteorological record

The Eureka Weather Station, operated by ECCC, has long-term records collected since 1947. In this work, Eureka hourly weather reports for 2010-2017 have been used to classify measured and modelled TCO data on the basis of weather conditions (http://climate.weather.gc.ca/). Details of the observing, recording, and reporting of weather conditions can be found in MANOBS (Meteorological Service of Canada, 2015). For example, when no weather or obstructions to visibility occur, weather conditions are reported as Clear (0 tenths), Mainly clear (1 to 4 tenths), Mostly cloudy (5 to 9 tenths), and Cloudy (10 tenths), based on the amount (in tenths) of cloud covering the dome of the sky.

## 2.6 Radiative Transfer Simulations

The radiative transfer model SCIATRAN (Rozanov et al., 2005, http://www.iup.uni-bremen.de/sciatran/) has been used to simulate the intensity of the scattered solar radiation observed on the ground. The model is designed to be used in any standard observation geometry (e.g., limb, nadir, zenith, or off-axis) by satellite, ground-based, or airborne instruments in ultraviolet, visible, and near-infrared spectral regions.

In this work, simulations of radiance have been performed for ground-based zenith-sky viewing observations in the visible band with varying aerosol and cloud optical depths. In the simulations, SCIATRAN standard trace gas volume mixing ratio ($O_3$, $NO_2$, $SO_2$ and etc.), pressure, and temperature profile scenarios are used, which are obtained from a 2-D chemical-dynamical model developed at the Max Planck Institute for Chemistry (MPIC, Brühl and Crutzen, 1993). Aerosol scattering is simulated by using the Henyey-Greenstein phase function with aerosol scenarios taken from LOWTRAN 7. Rayleigh scattering and ozone absorption are included. Different surface albedos (0.9 for winter conditions and 0.06 for summer conditions) are also assumed for different seasons.





## 3 Cloud screening

The cloud-screening algorithm has three steps and uses the calibrated CI, temporal smoothness of the CI, and temporal smoothness of $O_4$ dSCDs as proxies in cloud screening. In the first step, the measured CI is calibrated using a statistical method, and a threshold for clear-sky conditions is determined based on RTM simulations (described in below). Next, the

temporal smoothness of CI and $O_4$ dSCDs measured each day are labelled by a high-frequency filter (local regression method). Third, ozone dSCDs that passed the first two steps (identified as not cloud contaminated), are used in the so-called cloud-screen Langley plot method and converted to VCDs.

### 3.1 Colour index calibration

The CI is the ratio of the intensity of sunlight at two different wavelengths. For radiometrically calibrated instruments (such

as Brewer instruments and sun photometers), their measured intensity can be used as a good indication for sky condition (Fioletov et al., 2002, 2011). However, DOAS instruments are normally uncalibrated (Platt and Stutz, 2008) and their measured spectral intensity cannot be directly used to infer sky conditions (Gielen et al., 2014; Wagner et al., 2014, 2016; Wang et al., 2015). Wagner et al. (2016) proposed a statistical method to perform absolute calibration of the CI and $O_4$ measured by MAX-DOAS instruments. In the current work, following Wagner et al. (2016), an absolute calibration is

performed on ZS-DOAS CI, but the method is modified for use under high Arctic conditions.

The CI we use here is defined as the intensity ratio of two measured wavelengths (shorter to the longer wavelength). For example, UT-GBS spectra extend from about 350 to 560 nm, and intensities of 450 and 550 nm were selected to calculate the CI as:

$$CI = \frac{I_{450nm}}{I_{550nm}}. \qquad (1)$$

Other pairs of intensities proposed in other studies, such as 360/385, 360/550, 405/550, and 425/490 (Sarkissian et al., 1991; Wagner et al., 1998, 2014, 2016; Hendrick et al., 2011; Gielen et al., 2014) were all tested for UT-GBS. The 450/550 pair was found to be the most reliable one for the ZS-DOAS instruments used in this work.

As pointed out in previous studies (Gielen et al., 2014; Wagner et al., 2014, 2016), the zenith-sky CI measured in cloudy conditions is smaller than that in clear-sky condition. This is because the cloud enhances the scattering at the longer

wavelength due to enhanced Mie scattering. Figure 1 shows the measured CI from the UT-GBS in 2011. The plot is colour-coded by the density of the scattering points, and the coloured lines are examples of the CI simulated by the radiative transfer model under different sky conditions. Two distinct branches of the CI are revealed: the upper branch (measured CI value about 2) indicates clear sky conditions, while the lower branch (measured CI value about 1.2) indicates cloudy sky conditions. The CI can efficiently distinguish cloudy and clear conditions only when the SZA is smaller than about 85°; the

two CI branches merge at SZA close to 90°.

From Figure 1, it appears that the determination of a threshold to separate cloudy CI and clear-sky CI is straightforward. However, this type of CI density plot varies from instrument to instrument, and even from year to year (e.g., if the instrument




optics change). Thus the threshold is not a constant. To determine the threshold, the simple solution would be to compare the measured CI with RTM simulations. However, Figure 1 also shows a clear offset between the measured and simulated CI curves. For example, the lowest measured CI at SZA = 60° is about 1.3, while the RTM shows the lowest value could be about 1. Thus, the calibration of CI is necessary to correct this offset.

Following Wagner et al. (2016), the calibrated CI ($CI_{cal}$) is given by the multiplication of measured CI ($CI_{meas}$) by a constant factor β:

$$CI_{cal} = \beta \cdot CI_{meas} \ . \qquad (2)$$

To adapt the method of Wagner et al. (2016) (which is based on SZA < 55°) to high-latitude conditions, CI data with SZA < 85° are used in this work.

The process used to calibrate the data is illustrated in the example in Figure 2. First, we define a so-called cloudy envelope (see the red shaded area in Figure 2a) based on RTM simulations. The top of the cloudy envelope is defined as simulated CI with cloud optical depth (COD) = 1.5, whereas the bottom of the envelope is defined by the lowest simulated CI from all RTM simulations. Next, we assume the best estimated β should make most of CIs of the cloudy branch fall into this cloudy envelope (as shown in Fig. 2a as before calibration and Fig. 2b as after calibration), using the method described in the

paragraph below. Following Gielen et al. (2014), we also categorize the calibrated CI values into three regimes as shown in Figure 2b: (1) cloudy, when $CI_{cal}(SZA) < CI_{COD=1.5}$, (2) clear, when $CI_{cal}(SZA) > CI_{visibility=50km}$, and (3) intermediate, CI in between cloudy and clear condition.

Figure 3 shows examples of the estimation of β values for both UT-GBS and SAOZ in various years. For example, in Figure 3a, the percentage of measurements that fall into the cloudy branch envelope is shown by the purple line for various β

(partially hidden by the dashed green line), and the corresponding maximum is for β = 0.82. For quality control purposes, a Gaussian fitting (green dash line) is applied to the β estimation curve (the purple curve), which also gives a β(gauss) (the back vertical dash line) with 95% confidence bounds (the vertical solid green box). For years when there are enough cloudy measurements, the β(gauss) value is close to the estimated β value, indicating the good reliability of the calibration result for that year. The estimated β values for SAOZ were more stable than those for UT-GBS. This is because this SAOZ instrument

was almost untouched after it was first deployed at Eureka. However, the UT-GBS, as a travelling instrument, has been disassembled and reassembled several times over the eight years covered in this work.

In Figure 3, the blue, red, and yellow lines indicate the percentage of measurements categorized into those three sky-condition regimes (clear, intermediate, and cloudy). For UT-GBS 2011 measurements (Figure 3a), about 49% of measured spectra are labelled as clear, 14% as intermediate, and 37% as cloudy. In short, after this CI calibration, a CI sky condition

label (clear, intermediate, or cloudy) is generated for each spectrum. Spectra with CI sky condition labelled as cloudy can be filtered out.





## 3.2 Smoothness of CI and $O_4$ dSCDs

As shown in previous publications, the measured CI and $O_4$ dSCDs vary smoothly during the day if there are no rapidly changing clouds (Gielen et al., 2014; Wagner et al., 2014, 2016; Wang et al., 2015). Thus, the temporal smoothness of CI and $O_4$ dSCDs can be used as complementary sky condition labels. Details of how the smoothness of CI and $O_4$ dSCDs was

quantified are presented in Appendix A.

## 3.3 Langley plot method

Following Hendrick et al. (2011), the ozone dSCDs are converted to ozone VCDs (TCO) using the following equation,

$$VCD(SZA) = \frac{dSCD(SZA) + RCD}{AMF(SZA)} \qquad (3)$$

where the VCD, dSCD, and AMF are all functions of SZA. The reference column density (RCD) is the residual ozone

amount in the reference spectrum that is used in the DOAS analysis. The dSCD is directly obtained by DOAS analysis (using the QDOAS software). The AMF is extracted from the NDACC ozone AMF LUT, based on the latitude and elevation of the PEARL Ridge Lab, day of the year, sunrise or sunset conditions, wavelength, SZA, surface albedo, and ozone column (daily TCOs interpolated from daily or weekly Eureka ozonesonde data). The inclusion of ozonesonde data in the AMF calculations improves the results, especially under vortex conditions (Bassford et al., 2001). The RCD value is retrieved

using the so-called Langley plot method (Hendrick et al., 2011).

In general, by rearranging Equation (3), for each twilight period, a linear fitting of dSCDs versus AMFs is made, from which the RCD is given by the intercept value (AMF = 0). In this work, for each twilight, ozone dSCDs in the NDACC-recommended SZA range (86° to 91°) were selected, when those dSCDs were available. Otherwise, to adapt to the high-latitude condition, the nearest available 5° SZA range was used (Adams, 2012). For quality control purposes, any fit with

less than eight measurements or with a coefficient of determination ($R^2$) less than 0.9 was discarded.

For the UT-GBS, a daily average RCD was calculated from the morning and evening twilight RCDs because a daily reference spectrum (recorded at high sun around local noon) was used in the DOAS analysis. Applying this daily RCD in Equation (3), a group of VCDs (at different SZA) can be retrieved for that day. Next, sunrise and sunset VCDs were produced from the weighted mean of the VCD(SZA) (weighted by the DOAS fitting error divided by the AMF, Adams,

2012). These sunrise and sunset VCDs are the final product of ZS-DOAS TCO data, referred to as GBS data.

The difference between SAOZ and GBS TCO data processing is that SAOZ uses a fixed reference spectrum in its DOAS analysis. For SAOZ 2010-2017 observations, only three fixed reference spectra were used, from day 94 of the year 2010, day 126 of the year 2011, and day 101 of the year 2016. Thus, for SAOZ, three fixed RCDs were used for 2010 ($5.0 \times 10^{19}$ molec cm$^{-2}$), 2011 ($1.6 \times 10^{19}$ molec cm$^{-2}$), and 2012-2017 ($4.4 \times 10^{19}$ molec cm$^{-2}$) measurements. Other settings in the SAOZ TCO

retrieval (such as SZA range, quality control) are same as for the GBS data.





### 3.4 Cloud-screened Langley plot method

The cloud-screened Langley plot method is widely used for ground-based AOD measurements using sun photometers (Dayou et al., 2014). In general, this method is based on an objective cloud-screening algorithm, which is used to select cloudless data from a continuous time series that needed for the regression. With the information from the CI value label
(Section 3.1, assigned for spectra with SZA < 85°), and CI and $O_4$ smoothness labels (Section 3.2, assigned for spectra with SZA < 91°), we assigned a sky condition flag to each spectrum. If any of the three labels indicate cloudy conditions, the corresponding spectrum is flagged as cloudy, and it is excluded from the cloud-screened Langley plot. When cloud-affected spectra have been removed, the same criteria are applied to the cloud-screened Langley plot as apply for the conventional Langley plot (e.g., requires 9 data points and $R^2$ >0.9). The resulting cloud-screened GBS (SAOZ) TCO data is referred to as
$GBS_{CS}$ ($SAOZ_{CS}$). Table 1 summarizes the measured and modelled ozone data products.

### 4 Weather impacts and statistical uncertainty estimation

TCO time series (2010-2017) from all instruments and MERRA-2 are shown in Figure 4. In general, the seasonal cycles of the TCO from all ground-based instruments and the model track well with each other. The Brewer TCO has 3-5 minute temporal resolution; to pair with UT-GBS and SAOZ data, the Brewer TCO is resampled semi-daily by averaging data
collected for each half of the day. MERRA-2 TCO has a 3-hour temporal resolution, thus MERRA-2 TCO measured nearest in time with UT-GBS and SAOZ is used. The hourly weather records are resampled semi-daily by using the "median weather type" for each half of the day. For example, a weather condition (semi-daily) is cloudy, if most hourly weather records in that half day are cloudy. From 2010 to 2017, UT-GBS and Brewer had 960 coincident measurements, of which 182 coincident measurements were made in clear-sky conditions, and 102 coincident measurements were made in cloudy
conditions. Other major weather conditions for UT-GBS and Brewer coincident measurements include mainly clear (229), mostly cloudy (307), ice crystals (69), rain (11), and snow (38). Measurements made in other minor weather conditions such as blowing snow, fog, and rain showers only account for 2-3 % and are neglected.

### 4.1 Weather impacts on TCO accuracy

Without categorizing TCO measurements by weather conditions, the GBS dataset has a -0.23 ± 0.24 % mean bias compared
with Brewer, where the uncertainty is the standard error of the mean. Similarly, SAOZ has -0.16 ± 0.16 % mean bias relative to Brewer. This result is slightly better than Adams et al. (2012), who reported the mean relative difference between the GBS (SAOZ) and Brewer TCO measurements at Eureka as -1.4% (0.4%) for 2005-2011. These results (at Eureka) are better than the high-latitude agreement reported by Hendrick et al. (2011), who found that SAOZ TCO (1990-2008) were systematically lower than Brewer TCO at Sodankyla (67°N, 27°E) by 3-4 %, with the largest discrepancies in the spring and fall. Hendrick
et al. (2011) suggested that this bias was due to the temperature dependence (Kerr et al., 1988; Van Roozendael et al., 1998;





Kerr, 2002; Scarnato et al., 2009; Zhao et al., 2016b) and uncertainty in the ozone cross-section (Bass and Paur, 1985) used in Brewer measurements.

The agreement between the GBS, SAOZ, and Brewer in Adams et al. (2012) (and this study) is notable given the challenges of taking ZS-DOAS measurements at 80°N, particularly in the summer when measurements within the NDACC-recommended SZA range are not available. With help from the Eureka weather record, we can further explore the datasets and quantify the impact of weather, and improve our understanding of these comparison results.

In order to quantify the effects of weather on the ZS-DOAS data, coincident measurements were characterized according to the five major weather conditions from the Eureka weather record observations. Box plots for percent differences between the datasets were produced, as shown in Figure 5. Overall, the box plots demonstrate that biases between the ZS-DOAS and reference datasets are dependent on weather conditions. This is discussed in more detail below.

### 4.1.1 Weather impacts without the cloud-screen algorithm applied

The effect of weather on the GBS and SAOZ datasets is clear in the comparisons against the Brewer datasets (Figure 5a). For clear conditions, GBS (SAOZ) has $-0.30 \pm 0.54$ % ($-0.79 \pm 0.28$ %) mean bias; while for cloudy condition, this bias increases to $1.22 \pm 0.71$ % ($0.31 \pm 0.61$ %). Therefore, there is a 1.5 % (1.1 %) difference (statistically significant) between GBS (SAOZ) clear-sky measurements and cloudy-sky measurements; this difference is referred to as the clear-cloudy difference in the rest of this work.

This demonstrates that the good general agreement (low bias) between GBS (SAOZ) TCO and Brewer TCO reported in Section 4.1 is due to a combination of a negative bias in clear-sky conditions and a positive bias in cloudy conditions. Thus, if only clear-sky measurements are selected, ZS-DOAS measurements have a negative bias compared to Brewer measurements, which agrees with previous findings (Van Roozendael et al., 1998; Hendrick et al., 2011).

Measurements during other precipitation conditions (snow and rain) are relatively sparse (less than 50 coincident measurements, not shown here), since Brewer direct-sun measurements need a clear view toward the sun. The GBS TCO has a large negative bias ($-4.66 \pm 0.84$ %) in ice crystal conditions, while SAOZ TCO is almost unaffected ($-0.26 \pm 0.54$ %). One possible explanation for this discrepancy is that the UT-GBS has a much narrower field-of-view (0.2-2°) than SAOZ instruments (4°). However, with the limited coincident measurements, it is difficult to fully understand this feature.

To further study the impact of weather on ZS-DOAS TCO, we use a reference TCO dataset (other than Brewer) whose data quality is not affected by the weather. As described in Section 2.4, MERRA-2 TCO data has been used in previous studies, and agrees well with Brewer data at Eureka. Comparison results are shown in Figure 5b and d. There are approximately twice as many coincident measurements for MERRA-2 compared with Brewer.

Figure 5b shows that in clear conditions, GBS (SAOZ) has a $-2.92 \pm 0.42$ % ($-2.47 \pm 0.24$ %) mean bias; while in cloudy conditions, this bias shifted to a positive value, $2.35 \pm 0.46$ % ($2.21 \pm 0.44$ %). Therefore, the clear-cloudy difference for GBS (SAOZ) TCO is 5.3% (4.7%), and it is statistically significant. This difference is larger than the clear-cloudy difference relative to the Brewer TCO. This may be because there are more coincident data points with MERRA-2 in early spring (late





February to March); the ZS-DOAS TCO measurements in early spring are not as accurate as in late spring and early summer (late March to early May), mainly due to the lack of high sun reference spectra. Furthermore, Brewer has no measurements in heavy cloud conditions and so Brewer TCO may be clear-sky biased.

Using MERRA-2 sPV$_{490k}$, for the 2010-2017 period, 7.8 % (11.0 %) of UT-GBS (SAOZ) TCO measurements were made

when polar vortex was above Eureka. Measurements inside the polar vortex (not shown here) were filtered out to assess whether the location of the polar vortex relative to the instrument line-of-sight and model sampling is the cause of this large clear-cloudy difference. However, the clear-cloudy differences for both GBS and SAOZ are almost unchanged (5.4% for GBS, 5.0% for SAOZ). During clear conditions, GBS (SAOZ) has -2.95 ± 0.45 % (-2.78 ± 0.24 %) mean bias, while during cloudy conditions, the bias is 2.41 ± 0.47 % (2.25 ± 0.47 %).

**4.1.2 Weather impacts with cloud-screen algorithm applied**

Comparisons between the cloud-screened ZS-DOAS measurements and the reference datasets are also shown in Figure 5. This algorithm successfully filtered more of the measurements made when clouds had been observed at the Eureka Weather Station. For example, Figure 5c shows that the number of coincident measurements between SAOZ and Brewer decreased from 227 to 214 for clear conditions. For mostly cloudy conditions, this number decreased from 209 to 122. Note that this

algorithm is not designed to simply discard all TCO measurements made in cloudy days, but only to remove individual spectra that are cloud contaminated. For example, even for a cloudy day, if clouds cleared up during part of the twilight period, this algorithm may produce ZS-DOAS TCO data (if other criteria also met, described in Section 3.3 and 3.4).

Figure 5a shows that the GBS$_{CS}$ (SAOZ$_{CS}$) has a -2.10 ± 0.66 % (-1.09 ± 0.29 %) mean bias relative to Brewer, while during cloudy conditions, the bias is -0.50 ± 1.55 % (-0.56 ± 0.76 %). Therefore, the GBS$_{CS}$ (SAOZ$_{CS}$) data have a negative bias

compared to Brewer, even during cloudy conditions, which is expected for high quality cloud-free measurements (see Section 4.1.1). The clear-cloudy difference for GBS$_{CS}$ (SAOZ$_{CS}$) TCO is 1.6% (0.53%), which is not statistically significant, suggesting that a larger sample size is needed to infer this difference. Similarly, if MERRA-2 TCO data are used as the reference (see Figure 5b), during clear conditions, GBS$_{CS}$ (SAOZ$_{CS}$) has -3.76 ± 0.47 % (-2.56 ± 0.24 %) mean bias; while during cloudy conditions, the bias is 1.26 ± 0.66 % (2.33 ± 0.57 %).

The effectiveness of the cloud-screening algorithm is further demonstrated by scatter plots for Brewer versus GBS and GBS$_{CS,}$ shown in Figure 6 and 7. When data for all weather conditions are considered together (Figure 6a and Figure 7a), after applying the cloud-screen algorithm, the slope of the linear fit improved from 0.92 to 0.96, the intercept decreased from 28.97 DU to 22.82 DU, and R increased from 0.91 to 0.92. The effectiveness of the algorithm is most apparent for cloudy conditions (Figure 6c and Figure 7c), for which the slope of the linear fit improved from 0.91 to 1.00, the intercept decreased

from 27.98 DU to 0.93 DU, and R increased from 0.91 to 0.92, although the number of coincident measurements decreased from 102 to 33. Similar improvements can be found for other weather types, especially for most cloudy condition (Figure 6e and Figure 7e).





Correlations were also examined for other pairs of measurements, such as Brewer vs. SAOZ and SAOZ$_{CS}$, MERRA-2 vs. GBS and GBS$_{CS}$ etc. These results are summarized in Figure 8, which shows the correlation coefficients for all pairs of TCO datasets. Most pairs of data have R value greater than 0.9, and the R values are lager for the cloud-screened datasets (crosses) than for the unscreened (circles).

## 4.2 Statistical uncertainty estimation

In addition to the accuracy studied in Section 4.1, another important aspect of the TCO datasets is their precision. By comparing the same quantity retrieved from different remote sensing instruments, the random uncertainties can be characterized from the measurements themselves (Grubbs, 1948; Fioletov et al., 2006; Toohey and Strong, 2007; Zhao et al., 2016b). Following the method of Fioletov et al. (2006), briefly explained in Appendix B, a statistical uncertainty estimation model is used to estimate random uncertainties for ZS-DOAS instruments (UT-GBS and SAOZ).

Figure 9a shows the resulting estimated random uncertainties. The first blue column on Figure 9a represents the estimated random uncertainty for GBS TCO, when using Brewer TCO as reference (see description in Appendix B). The number of coincident measurements is shown in Figure 9b. In general, GBS (SAOZ) has a random uncertainty of 4.04 ± 0.21 % (3.19 ± 0.17 %), when using the Brewer as the reference. If MERRA-2 is used as a reference, the random uncertainty for GBS and SAOZ is 3.86 ± 0.11 % and 2.80 ± 0.09 %, respectively. Thus, SAOZ TCO has about 1 % lower random uncertainty than GBS TCO. The estimated random uncertainties for GBS and SAOZ are both lower than 4.7 %, the precision value reported by Hendrick et al. (2011).

Theoretically, the cloud-screened TCO datasets (GBS$_{CS}$ and SAOZ$_{CS}$) should have lower random uncertainties than the conventional TCO datasets (GBS and SAOZ). The GBS$_{CS}$ (SAOZ$_{CS}$) has random uncertainty 3.86 ± 0.29 % (2.94 ± 0.19 %), when using Brewer as the reference. With MERRA-2 as the reference, GBS$_{CS}$ (SAOZ$_{CS}$) has a random uncertainty of 3.30 ± 0.11 % (2.64 ± 0.10 %). Although GBS$_{CS}$ and SAOZ$_{CS}$ have lower random uncertainties compared to GBS and SAOZ. The only significant improvement on random uncertainty is for UT-GBS, which decreased from 3.86 ± 0.11 % to 3.30 ± 0.11 % (red bar on GBS and GBS$_{CS}$ columns), when using MERRA-2 as reference. This improvement is most likely significant because the sample size is sufficient (2370 coincident measurements, see Figure 9b).

To further illustrate the cloud effect, the Eureka weather record is used as an extra filter to strength the cloud filtering. Measurements are preserved and used in uncertainty estimation only if it were made in clear or mostly clear recorded weather conditions. The yellow and green symbols represent the precision calculated with this extra filter applied. Filtering out all measurements made in non-ideal weather, the GBS random uncertainty improved from 4.04 ± 0.21 % to 2.78 ± 0.29 %, when using Brewer as reference (see the blue and yellow bars on the GBS column). For SAOZ, random uncertainty improved from 3.19 ± 0.17 % to 2.60 ± 0.26 % (blue and yellow bars on the SAOZ column). These improvements are both statistically significant. This result is close to the uncertainty budget table in Hendrick et al. (2011), in which ZS-DOAS TCO total precision can be improved by about 1 % in cloud-free conditions.



## 5 Conclusions

Clouds are the largest source of random uncertainty in ZS-DOAS TCO retrievals. This work provides a measurement-based evaluation of the effect of cloud conditions on ZS-DOAS TCO. A cloud-screening algorithm was developed to improve TCO data quality under cloudy conditions, one which could potentially be applied to the NDACC UV-visible network. With ozone measurements, weather observations, and models, this study helps answer the following questions.

- What is the effect of cloudy sky conditions on ZS-DOAS TCO data?

For the Eureka datasets, there is a statistically significant difference of 1-5 % between ZS-DOAS TCO measurements made under cloudy and clear-sky conditions.

- It has been estimated that clouds contribute 3.3% random uncertainty in the NDACC ZS-DOAS TCO retrieval (Hendrick et al., 2011). Thus by removing the cloud term from the error budget, ZS-DOAS TCO datasets should have their random uncertainty lowered by 1.3%. Can this value be verified by field measurements?

After removing cloudy measurements using weather records, the Eureka ZS-DOAS TCO random uncertainties are reduced by 0.6-1.3 %. Note that the 3.3% random uncertainty in Hendrick et al. (2011) is an upper limit of the impact of clouds on ozone AMFs, since it is based on sensitivity tests using parameter values for rather large stratus cloud (Shettle, 1989). Thus, the findings in this work agree with the results in Hendrick et al. (2011).

- TCO measurements in the high Arctic are challenging (e.g., low sun and large SZA in early spring). What is the general quality of ZS-DOAS TCO measured at Eureka?

Using a statistical uncertainty estimation model, TCO datasets from two ZS-DOAS instruments located at Eureka have been evaluated. GBS TCO has a random uncertainty of 3.9-4.0 %, while SAOZ TCO has a random uncertainty of 2.8-3.2 %. Both instruments have random uncertainties that are lower than the 4.7 % reported by Hendrick et al. (2011).

- Adams et al. (2012) and this work both found that the mean relative difference between the ZS-DOAS and Brewer TCO measurements at Eureka (e.g., 0.4 % for SAOZ 2005-2011, in Adams et al. (2012)) is better than the high-latitude agreement reported by Hendrick et al. (2011), who found a negative bias of 3-4% in SAOZ TCO (1990-2008) compared with Brewer TCO at Sodankyla (67°N). Given the challenges of taking ZS-DOAS measurements in the high Arctic (Eureka, 80°N), why do measurements in Eureka have such good agreement with Brewer data?

This good agreement is a combination of positive bias during cloudy conditions and a negative bias during clear conditions. For measurements under clear conditions only, GBS (SAOZ) has -0.30 ± 0.54 % (-0.79 ± 0.28 %) mean bias; while for cloudy conditions only, this bias is positive 1.22 ± 0.71 % (0.31 ± 0.61 %). However, if Brewer TCO is replaced by MERRA-2 TCO, during clear conditions, GBS (SAOZ) has -2.92 ± 0.42 % (-2.47 ± 0.24 %) mean bias; while during cloudy conditions, this bias is 2.35 ± 0.46 % (2.21 ± 0.44 %). In addition, in the high Arctic, Brewer TCO measurements are only available for a relatively short periods (from April to September), and thus the temperature effect (seasonal bias) in the Brewer TCO dataset is smaller compared to datasets collected at mid- and low-latitudes (Zhao et al., 2016b). Thus, it is likely the good agreement between ZS-DOAS and Brewer at Eureka is due to a combination of temperature, cloud, and other



effect. Answering this type of question about consistency between datasets is important for the NDACC UV-visible network to provide globally harmonized ZS-DOAS TCO datasets.

In addition to answering the scientific questions above, this work also provides the following contributions to ZS-DOAS measurements and data processing. (1) A cloud-screen algorithm for ZS-DOAS ozone measurements at high-latitude sites

has been developed. This algorithm can be modified and applied to low- and mid-latitude ZS-DOAS measurements. (2) Cloud-screened long-term (2010-2017) ZS-DOAS TCO datasets in Eureka have been produced, implementing the latest NDACC UV-visible network ozone retrieval protocol. These TCO datasets will be used for validation of space-based ozone measurements by the Optical Spectrograph and Infra-Red Imager System (OSIRIS) and the Atmospheric Chemistry Experiment (ACE) in a future paper.

**Acknowledgments**

The UT-GBS data are available from the Canadian Network for the Detection of Atmospheric Change (CANDAC, www.candac.ca) and the Network for the Detection of Atmospheric Composition Change (NDACC, https://ftp.cpc.ncep.noaa.gov/ndacc/station/eureka). SAOZ data are available from the Canadian Arctic ACE/OSIRIS Validation Campaign (eureka.physics.utoronto.ca). Brewer and ozonesonde data are available from the World Ozone and

UV Data Centre (https://doi.org/10.14287/10000001). Eureka weather station records are available from Environment and Natural Resources in Canada (www.canada.ca/en/services/environment.html). Any additional data may be obtained from Xiaoyi Zhao (xizhao@atmosp.physics.utoronto.ca).

CANDAC/PEARL funding partners are the Arctic Research Infrastructure Fund, Atlantic Innovation Fund/Nova Scotia Research Innovation Trust, Canadian Foundation for Climate and Atmospheric Science, Canada Foundation for Innovation,

Canadian Space Agency (CSA), Environment and Climate Change Canada (ECCC), Government of Canada International Polar Year, Natural Sciences and Engineering Research Council (NSERC), Ontario Innovation Trust, Ontario Research Fund, Indian and Northern Affairs Canada, and the Polar Continental Shelf Program. The spring 2010-2017 measurements were also supported by the Canadian Arctic ACE/OSIRIS Validation Campaigns funded by CSA, ECCC, NSERC, and the Northern Scientific Training Program. MERRA-2 work at the Jet Propulsion Laboratory, California Institute of Technology,

was done under contract with the National Aeronautics and Space Administration. Xiaoyi Zhao was supported by the NSERC CREATE Training Program in Arctic Atmospheric Science, the Probing Atmosphere in the High Arctic (PAHA) project, and the Visiting Fellowships in Canadian Government Laboratories program. We thank CANDAC/PEARL/PAHA PI James Drummond, ACE Validation Campaign PI Kaley Walker, PEARL Site Manager Pierre Fogal, the CANDAC operators, and the staff at ECCC's Eureka Weather Station for their contributions to data acquisition, and logistical and on-

site support. We thank Jonathan Davies and David Tarasick from ECCC for providing ozonesonde data. We thank Thomas Danckaert, Caroline Fayt, Michel Van Roozendael, and others from IASB-BIRA for providing QDOAS software and



NDACC UV-visible working group for providing NDACC UV-visible ozone AMF LUT. We thank Alexei Rozanov from IUP Bremen for providing the SCIATRAN radiative transfer model.

**Appendices**

### A. Smoothness of CI and $O_4$ dSCDs

To determine the smoothness of the calibrated colour index, following Gielen et al. (2014) we define a temporal smoothness label (TSL) for CI as:

$$TSL_{CI} = \left| \left\{ \frac{[CI_{cal}(t) - LOWESS(t,x)]}{LOWESS(t,x)} \right\} \right| \qquad (4)$$

where $t$ is local time, $LOWESS(t,x)$ is the fitted daily CI curve using the locally weighted scatterplot smoothing (LOWESS fit, based on local least-squares fitting applied to a specified x fraction of the data) (Cleveland and Devlin, 1988). The local fitting fraction $x$ is selected as 50%. Only CIs measured with SZA < 92° are used in the daily curve fitting. If $TSL_{CI} > 0.1$, we label the spectrum as cloudy (refer to as CI smoothness label).

Similarly, we define a TSL for $O_4$ absorption as:

$$TSL_{O4} = \left| \left\{ \frac{[dSCD_{O4}(t) - LOWESS(t,x)]}{LOWESS(t,x)} \right\} \right| \qquad (5)$$

where $t$ is local time, $LOWESS(t,x)$ is the fitted daily $O_4$ dSCDs using LOWESS fit, and the local fitting fraction $x$ is selected as 50%. Only $O_4$ dSCDs measured with SZA < 92° are used in the daily curve fitting. If $TSL_{O4} > 0.2$, we label the spectrum as cloudy (referred to as the $O_4$ smoothness label).

These thresholds for $TSL_{O4}$ and $TSL_{CI}$ both follow Gielen et al. (2014), but instead of using LOWESS fit, they used a double sine function to simulate the low-frequency variation of CI and $O_4$.

### B. Statistical uncertainty estimation

Random uncertainties for ZS-DOAS measurements can be determined using a statistical estimation method (Grubbs, 1948; Fioletov et al., 2006; Toohey and Strong, 2007; Zhao et al., 2016b). As an example, we define the two types of measured TCO (denoted as $M_1$ and $M_2$, for Brewer and ZS-DOAS measurements, respectively) as simple linear functions of the true TCO value ($X$) and instrument random uncertainties ($\delta_1$ and $\delta_2$), and assume that there is no multiplicative or additive bias between ZS-DOAS and Brewer, giving

$$M_1 = X + \delta_1$$
$$M_2 = X + \delta_2 . \qquad (6)$$

If we assume that the instrument random uncertainties are independent of the measured TCO, the variance of $M$ is the sum of the variances of $X$ (around the mean of the dataset) and $\delta$,

$$\sigma_{M_1}^2 = \sigma_X^2 + \sigma_{\delta_1}^2$$





$$\sigma_{M_2}^2 = \sigma_X^2 + \sigma_{\delta_2}^2 \ . \qquad (7)$$

If the difference between ZS-DOAS and Brewer does not depend on $X$ (no multiplicative bias), and the random uncertainties of the two instruments are not correlated, then the variance of the difference is equal to the sum of the variance of the random uncertainties,

$$\sigma_{M_1 - M_2}^2 = \sigma_{\delta_1}^2 + \sigma_{\delta_2}^2 \ . \qquad (8)$$

Since we have the measured TCO and the difference between the ZS-DOAS and Brewer datasets, the variance of the instrument random uncertainties can be solved by

$$\sigma_{\delta_1}^2 = \left(\sigma_{M_1}^2 - \sigma_{M_2}^2 + \sigma_{M_1 - M_2}^2\right)/2$$
$$\sigma_{\delta_2}^2 = \left(\sigma_{M_2}^2 - \sigma_{M_1}^2 + \sigma_{M_1 - M_2}^2\right)/2 \ . \quad (9)$$

Equation (6) can be used to estimate the standard deviation of instrument random uncertainties ($\sigma_{\delta_1}$ and $\sigma_{\delta_2}$). The variances $\sigma_{M_i}^2$ and $\sigma_{M_1 - M_2}^2$ can be estimated from the available measurements (with some uncertainty). The uncertainties in the $\sigma_{\delta_1}^2$ and $\sigma_{\delta_2}^2$ estimates depend on the sum of all three variances $\sigma_{M_1}^2$, $\sigma_{M_2}^2$ , and $\sigma_{M_1 - M_2}^2$, and can be high even if the estimated variance itself is low (but one or more of the variances $\sigma_{M_1}^2$, $\sigma_{M_2}^2$ , and $\sigma_{M_1 - M_2}^2$ are high). Thus, the estimates are only as accurate as the least accurate of these parameters. Following the method in Zhao et al. (2016b), the variance estimates can be improved by

increasing the number of data points or by reducing variances of $X$ by removing some of its natural variability. Thus, the $M_1$ and $M_2$ used in the statistical uncertainty estimation are replaced by so-called residual ozone, which is defined as the difference between the semi-daily measured TCO and its weekly mean.

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





**Table 1.** Summary of measured and model ozone data products.

| Instrument/Model | Total Column Ozone Data (Abbreviation) | Observation Geometry | Solar Zenith Angle | Use Daily Reference Spectrum | Use Cloud Screening Algorithm |
|---|---|---|---|---|---|
| UT-GBS | GBS | Zenith-Sky | 86-91°* | Yes | No |
| | GBS$_{CS}$ | Zenith-Sky | 86-91°* | Yes | Yes |
| SAOZ no. 7 | SAOZ | Zenith-Sky | 86-91°* | No | No |
| | SAOZ$_{CS}$ | Zenith-Sky | 86-91°* | No | Yes |
| Brewer no. 69 | Brewer | Direct-Sun | < 80° | N/A | N/A |
| MERRA-2 | MERRA-2 | N/A (atmospheric reanalyses) | | | |

\* At Eureka, this NDACC-recommended SZA range is available for only two months in a year. Thus to adapt to the high-latitude conditions, the nearest available 5° SZA range was used when necessary.





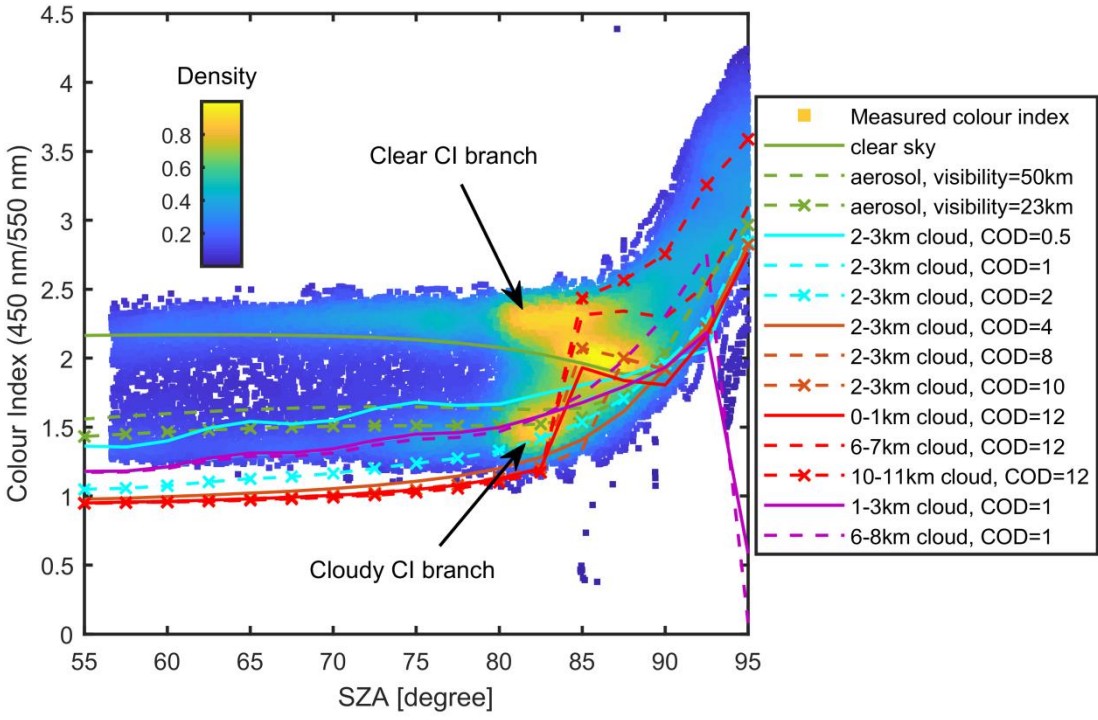

**Figure 1.** Colour index as a function of solar zenith angle. The measurements are from the UT-GBS in 2011, colour-coded by the normalized density of the points. Colour lines are examples of radiative transfer model CI simulations, using a surface albedo of 0.06 and the MPIC climatology ozone profile. Cloud height and cloud optical density (COD) indicated in the legend.





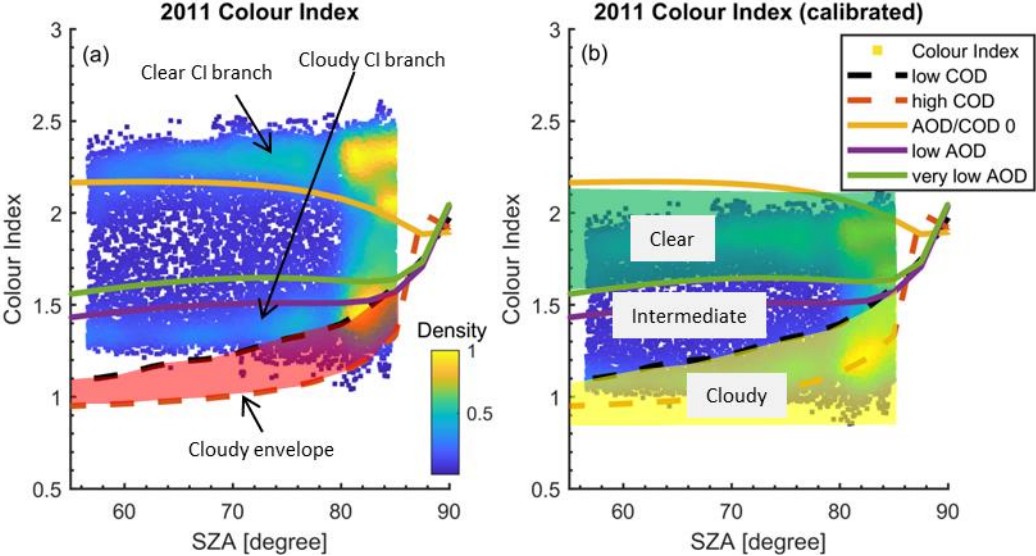

**Figure 2.** UT-GBS 2011 colour index before and after calibration, colour-coded by the normalized density of the points. Colour lines are SCIATRAN radiative transfer model CI results, with cloud optical depth (COD) and aerosol optical depth (AOD) conditions indicated in the legend. Panel (a) shows the measured CI and panel (b) shows the calibrated CI. Note that any measurements with solar zenith angle (SZA) > 85° have been removed in this calibration process.




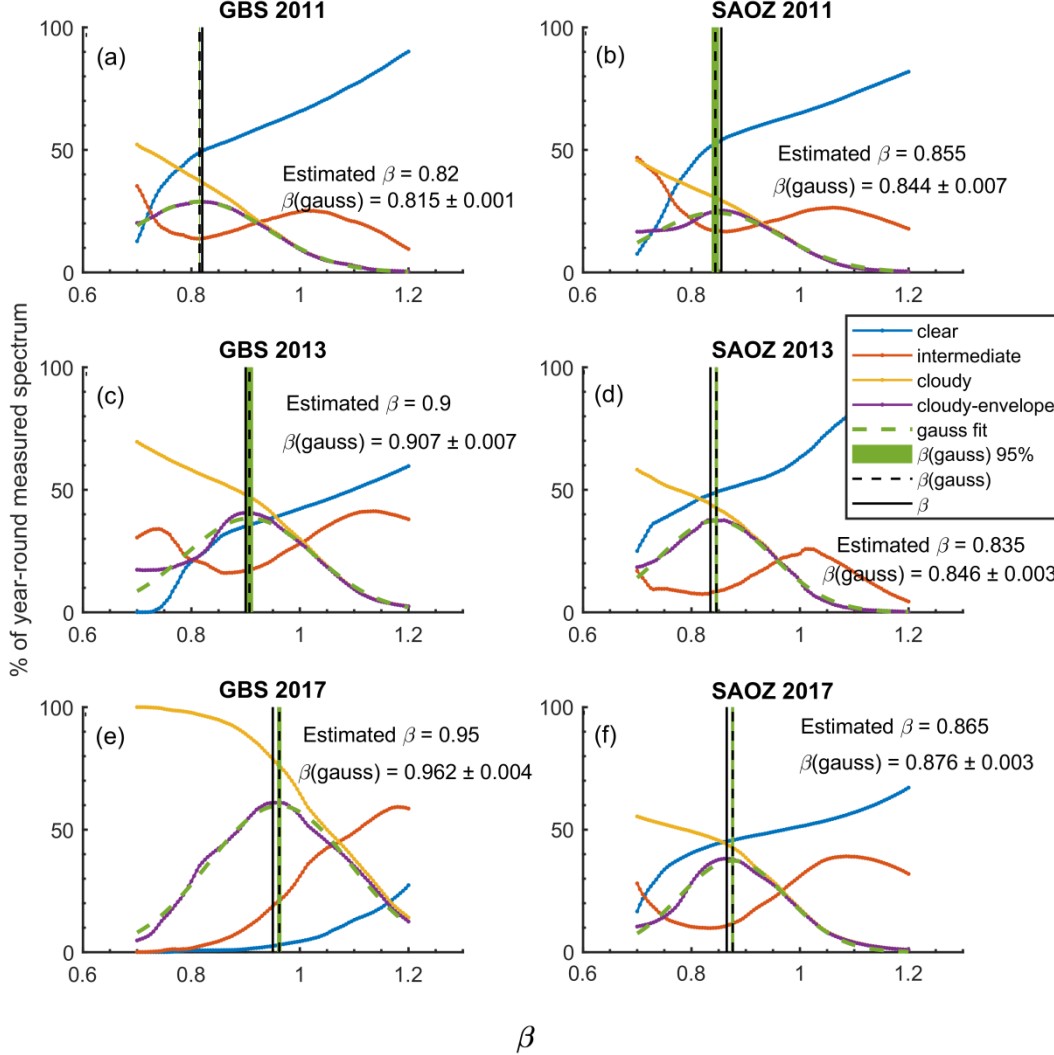

**Figure 3.** Examples of colour index calibration factor (β) determination. The y-axis is the percentage of year-round measured spectra, the x-axis is the β value used in the calibration. Solid lines represent measurements under different weather categories (blue for clear, red for intermediate, yellow for cloudy, and purple for cloudy envelope). Estimated β and β(gauss) values are shown by vertical solid black line and green dashed line, respectively. The vertical green shaded area is the 95% confidence bound of the β(gauss) value. Instrument name and measurement year are indicated on each panel. Note that any measurements with solar zenith angle (SZA) > 85° have been removed in this calibration process.



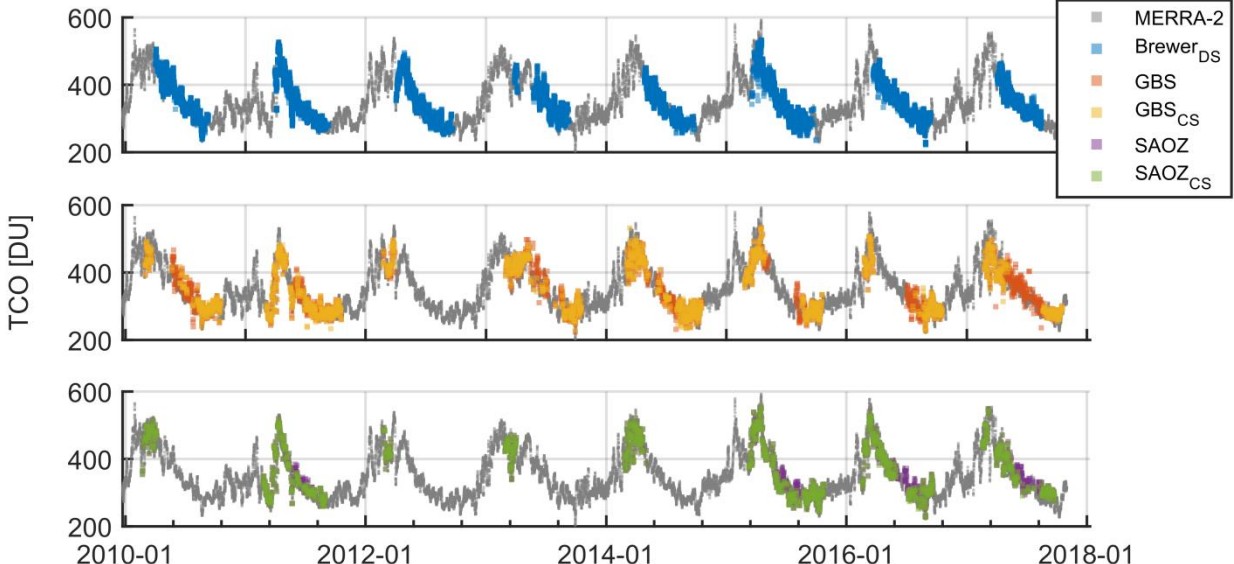

**Figure 4.** Time series of measured and modelled total column ozone (TCO) at Eureka.



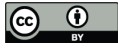

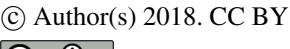

**Figure 5.** Weather impacts on total column ozone measurements: (a) percentage difference between UT-GBS and SAOZ TCO with Brewer TCO, (b) percentage difference between UT-GBS and SAOZ TCO with MERRA-2 TCO. (c) number of coincident measurements corresponding to (a), (d) number of coincident measurements corresponding to (b). Different colours represent different datasets, as indicated in the legend. In (a) and (b), the hollow box represents the 75[th] to 25[th] percentile of the dataset, the target symbol (black dot with coloured circle around) represents the median value, the solid bar represents the mean value, and the error bars represent the standard error on the mean. In all panels, the x-axis represents weather types reported at the Eureka Weather Station.



**Figure 6.** Scatter plots of Brewer total column ozone vs. UT-GBS (GBS) TCO. Panel (a) shows the scatter plot of all coincident measurements of Brewer and GBS. Panels (b) to (h) show scatter plots with weather conditions indicated in their titles. On each scatter plot, the red line is the linear fit with intercept set to 0, the blue line is a simple linear fit, and the black line is the one-to-one line.





**Figure 7.** Scatter plots of Brewer total column ozone vs. UT-GBS cloud-screened TCO (GBS$_{CS}$). Panel (a) shows the scatter plot of all coincident measurements of Brewer and GBS$_{CS}$. Panel (b) to (h) show scatter plots with weather conditions indicated in their titles. On each scatter plot, the red line is the linear fit with intercept set to 0, the blue line is a simple linear fit, and the black line is the one-to-one line.





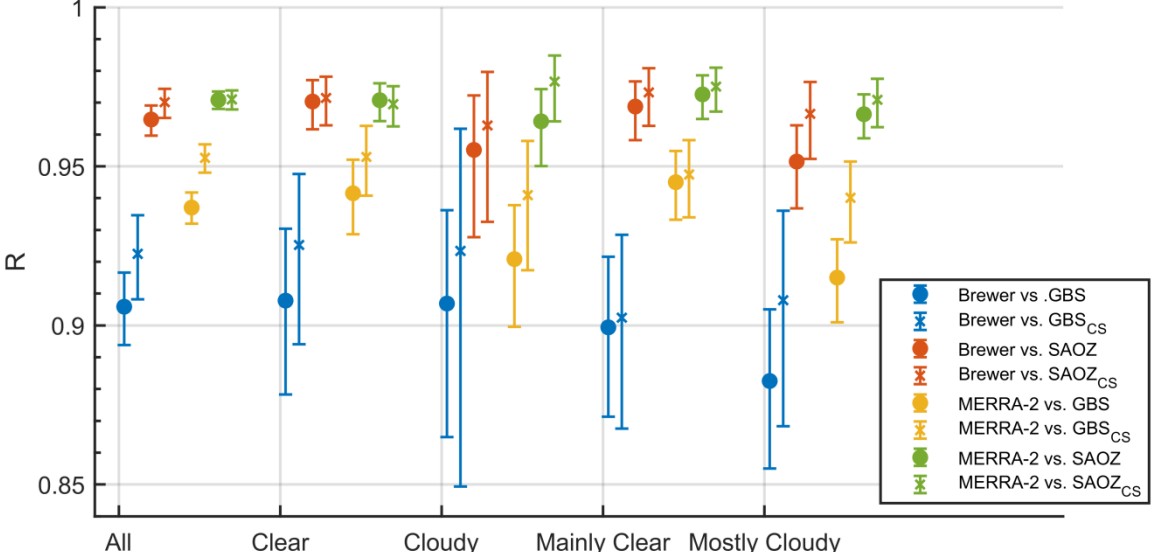

**Figure 8.** Correlation coefficients (R) of pairs of measured and modelled total column ozone datasets. The comparisons with GBS or SAOZ TCO datasets are shown by circles, and those with cloud-screened TCO datasets (GBS$_{CS}$ or SAOZ$_{CS}$) are shown by crosses. The error bars are the 95% confidence interval for each coefficient.



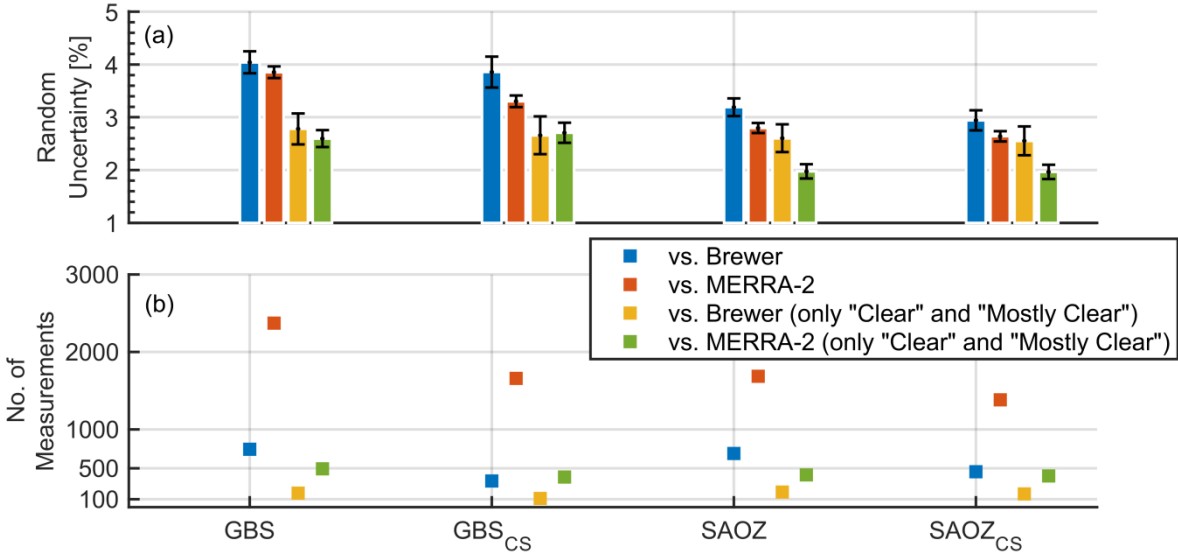

**Figure 9.** Statistical uncertainty estimation results. Panel (a) shows the estimated random uncertainties (%) and panel (b) shows the number of coincident measurements used. The x-axis indicates names of TCO datasets that been assessed. Colours represents different reference datasets (shown in legend).