# Peer review of "Assessing the Impact of Clouds on Ground-based UV-visible Total Column Ozone Measurements in the High Arctic"

_Atmospheric Measurement Techniques, 2018_

## Referee Comment (RC1) · Anonymous Referee #1 · 2 Oct 2018

The paper by Zhao et al. presents a cloud detection method for ground based zenith sky DOAS measurements. One particular advantage of this method is that it can be applied to measurements at high latitudes, for which the range of SZA is typically limited. The authors then use their cloud classification results to test the impact of clouds on measurements of the total ozone column. The cloud effect is quantified by comparison to a) direct sun Brewer/Dobson measurements and b) to model results. The authors find that excluding cloudy data reduces the uncertainties of the ozone measurements. This paper can be a potentially important paper for many groups performing and analysing zenith sky DOAS measurements. However, in its current form, many aspects are not (well) addressed, and also several findings need further clarification.

[Figure]

The paper needs major revisions, which are detailed below.

Major points:

A) I think the logic of the paper needs to be changed. In its current form the authors develop their cloud detection method and then directly apply it for the selection of the ozone measurements. In addition they use meteorological observations of cloud properties for the further assessment of the cloud effects on the ozone measurements. In my opinion, the first logical step after the development of the cloud classification algorithm would, however, be to compare the results of the cloud classification algorithm to independent cloud observations (e.g. meteorological observations) in order to validate the new algorithm. After successful validation, the algorithm could be applied to the ozone measurements.

B) The choice of the wavelength pair for the calculation of the CI is not be well justified. The authors write 'The 450/550 pair was found to be the most reliable one for the ZS-DOAS instruments used in this work'. It should be made clear in which respect the new choice is better over the other suggested wavelength pairs. (and what is meant with 'reliable'?) In my opinion the new pair is probably even problematic, because the measured radiance at 550 nm strongly depends on the ozone content (at high SZA). Assuming e.g. an ozone VCD of 300 DU and an AMF of 10, this results in an ozone optical depth at 550 nm of about 0.28. This can have a significant effect on the CI and makes the CI dependent on the ozone amount.

C) The effect of the surface albedo is not discussed. I would expect that it has a systematic influence on the CI. At high latitude sites the surface albedo changes strongly over the year. The authors should discuss this effect and should explain how they deal with the variability of the surface albedo.

D) The SZA dependence (e.g. in Fig. 2) of the model simulations and the measurements is very different (especially for the minimum values). The authors should discuss possible reasons for these differences (maybe related to change of albedo during the

year)? Also information on the input for the model simulations should be given, especially the ozone VDD and the surface albedo used for the simulations. In Fig. 1 several jumps are seen for the simulation results. What is the reason for these jumps? They seem to be not realistic.

E) In Fig. 1 it is seen that high clouds can have very similar CI as clear sky observations. The authors should check if this result is reasonable. If this simulation results are correct, I have some doubts about the ability of the algorithm to detect high clouds. These clouds might have a considerable effect on the ozone measurements.

F) The authors skip individual measurements, which are indicated as cloudy. I am not sure if this is a good procedure, because it leads to a variable selection of measurements (different numbers, different SZAs), which can have a systematic effect on the derived average ozone results. Also, if only a small number of measurements remains, the total uncertainty might increase. The authors should investigate how the selection of measurements affects the derived average O3 VCD. What is the minimum number of required measurements in a sequence? There is another, related point: it is written, that in some cases the SZA range of the selected measurements is shifted from the standard range (86-91°). How large is the maximum shift of the SZA range? For which situations is a shift applied? How does the shift affect the ozone results?

G) The effect of instrument degradation should be addressed. The authors write that in particular the differences in the calibration for the GBS instrument might be related to instrumental changes. The occurrence and strength of changes in the instrumental properties should be stated. Also gradual long term degradation should be investigated.

Minor points:

1) Can the authors explain, for which atmospheric conditions measurements fall into the category 'intermediate'?

2) In several parts of the paper, the cloud effects are referred to as 'random', e.g. in the abstract. In other parts, e.g. on page 3, line 17 it is stated that 'This leads to a random uncertainty of 3.3% for TCO calculated using the NDACC ozone AMF LUT between 86-91° SZA.' Then in the next sentence it is written 'In fact, clouds are the largest source of random uncertainty in ZS TCO.' In my opinion, cloud effects are systematic. Of course, depending on the cloud type, they might have different effects on the derived O3 VCD. Thus they can indeed introduce a random component. The authors should discuss these aspects in more detail. They also should be clear whether cloud effects are random or/and systematic.

3) In Fig. 3 the fitted curve seems to be not a pure Gaussian. Please provide details of the applied fit function.

4) Fig. 3: which SZA are included in these results?

5) Title: maybe add 'ground based' between 'on' and 'UV'?

6) Introduction: on page 3, lines 9-10, also the following reference might be included:

Erle F., Pfeilsticker K., Platt U, On the influence of tropospheric clouds on zenith scattered light measurements of stratospheric species, Geophys. Res. Lett., 22 , 2725-2728, 1995.

7) On page 5 it is written: 'Due to the decreased resolution at the edge of CCD, the ozone differential slant column densities (dSCDs) were retrieved in the 450-545 nm window, instead of the NDACC recommended 450-550 nm window.'

The Chappuis ozone absorption has no fine spectral structures. Is a high spectral resolution really needed for the ozone analysis in the visible? Maybe the NDACC window can still be used?

8) Fig. 2: Why have measurements for SZA > 85° been removed?

9) Section 3.2: How do the results based on the temporal variation agree with the

results derived from the CI threshold method?

10) Section 3.3: It is written that 'The inclusion of ozonesonde data in the AMF calculations improves the results, especially under vortex conditions (Bassford et al., 2001).'

This statement is unclear to me. Is the use of ozone sonde data an addition to the existing NDACC LUT? Is the original NDACC LUT used in this study or and updated LUT?

11) Page 14, line 18: It is written: 'Theoretically, the cloud-screened TCO datasets (GBSCS and SAOZCS) should have lower random uncertainties than the conventional TCO datasets (GBS and SAOZ).' I am not sure about this statement. One general effect of the cloud filter is that it removes measurements of a sequence. Thus the information content should be smaller than for a complete sequence. Also the selection of measurements becomes variable: e.g. on some days measurements for small SZA, and on other days large SZA might be filtered. This will lead to different biases and probably to an increased 'random' uncertainty.

---

## Referee Comment (RC2) · Anonymous Referee #3 · 25 Oct 2018

This work applies an algorithm to classify ZS-DOAS measurements at twilight according to cloud condition to understand the effect of clouds on the retrieval of total ozone column. This algorithm takes into account the colour index (CI), and the variation of CI and O4 measurements along the day to differentiate between three cloud scenarios: cloudy, clear and intermediate. The algorithm is applied to ZS-DOAS measurements for two different DOAS instruments located in a high latitude observatory at 80°N. The effect of clouds in DOAS TCO is investigated by comparison to Brewer TCO data series at the same observatory and to MERRA-2 data. General comments: I think the long data series at Eureka observatory is highly valuable due to the difficulty to keep instruments measuring with high quality in remote locations for so long. I think this

work can be very interesting for the interpretation of DOAS TCO measurements, and to assess the way that clouds, even low clouds can affect to DOAS measurements in the visible region. In my opinion this work should be published after addressing some mayor, minor and technical issues.

Major issues: (1) To assign the label of cloudy, clear or intermediate, the variation of O4 along the day is taken into account. I think this criterion can be stricter for GBS instrument than for SAOZ instrument due to their differences in FOV. As can be seen in figure 4 (although please, see technical comments about this figure), it seems that there are more "clear" data in the case of SAOZ than in the case of GBS. I was wondering if this fact could be due to the O4 criterion. In figure 3, it is quite surprising that for year 2011 clear, cloud and intermediate cases are quite close for both instruments but this situation changes considerably for 2013 and it is clearly different for 2017. But both instruments are located in the same observatory, how is possible that the number of clear/cloudy days in 2017 can be that different? Maybe the O4 criterion is too permissive for SAOZ and too strict for GBS? This could also have an effect in the difference on the bias for both instruments when compared to Brewer. If the algorithm is not properly working for SAOZ, some clear days can, in fact, be affected by clouds and that would explain the better agreement between SAOZCS and Brewer than GBSCS and Brewer.

(2) To be sure that the effect you observe in the bias when CS is applied to GBS TCO is only due to the presence of clouds, have you take into account that most of cloudy days happen out of the summer? What SZA do you use to calculate TCO at summer? Could the observed bias to Brewer have some to do with the major weight of summer days when you eliminate the cloudy days?

Minor issues: (1) Due to the high latitude of the observatory it is not possible to have DOAS measurements along the entire year. Please, in the description of the instrument include what is the annual period of measurements. From figure 4 and from data along the text it seems that the period is late winter to late autumn? It would be nice to know

the months when DOAS and Brewer can measure.

(2) Section 4.1. Why the current agreement to Brewer and GBS is better in this work than in the previous work by Adams et al.?

(3) Taking into account the current results, it seems that in the case of Hendrick et al., not all the observed discrepancies between DOAS and Brewer were due to the temperature dependence of XS used in Brewer analysis or in this work the Brewer analysis takes into account this dependence? Do you observe also the same seasonal difference (taking into account that you cannot observe the entire spring and fall at 80°N) that Hendrick et al. in the bias against the Brewer?

(4) Section 4.1.1, please indicate at any part of the text that the weather classification used here and in figure 5 is made by using meteorological data. If not, it is a little confusing.

Technical issues:

(1) Figure 4. Please, unify ticks in the horizontal axis. The lower graph is different from the previous ones and this makes very difficult to see properly the measurement periods. Grid in the middle of each year would be also very helpful. Colours in the legend are not coincident with the ones in the graphs. As GBSCS or SAOZCS are over imposed to GBS and SAOZ respectively, it seems that there are more data for the CS filtered data than without any filter. This is a little bit confusing at first, I am not sure that it can be addressed, maybe using hollow symbols for CS case? If possible it would be nice a greater graph.

(2) Sometimes the DOAS instrument GBS is called UT-GBS, please unify nomenclature along the text.

Please also note the supplement to this comment:
https://www.atmos-meas-tech-discuss.net/amt-2018-261/amt-2018-261-RC2-

supplement.pdf

---

## Author Comment (AC1) · 10 Jan 2019

**Response to Referee #1:**

Thank you to referee #1 for the helpful comments. Our responses are given below in black with the comments in blue. The new text in the modified manuscript is given in red (italicized).

**Referee #1:**

Major points:

A) I think the logic of the paper needs to be changed. In its current form the authors develop their cloud detection method and then directly apply it for the selection of the ozone measurements. In addition they use meteorological observations of cloud properties for the further assessment of the cloud effects on the ozone measurements. In my opinion, the first logical step after the development of the cloud classification algorithm would, however, be to compare the results of the cloud classification algorithm to independent cloud observations (e.g. meteorological observations) in order to validate the new algorithm. After successful validation, the algorithm could be applied to the ozone measurements.

Many previous researchers have already validated the methods of using CI and/or $O_4$ to identify clouds. The purpose of this work is to adapt those UV-vis cloud identification algorithms to Arctic conditions. The only available independent cloud observations for this study were meteorological observations. However, the meteorological observations are not an ideal dataset for the type of validation work because: 1) the frequency of meteorological observations is low (hourly) and 2) the meteorological observations do not include details such as cloud distribution or thickness of clouds. The meteorological observations were made based on the manual of surface weather observations MANOBS, published by Meteorological Service of Canada (Meteorological Service of Canada, 2015). For example, weather conditions are reported based on the amount of cloud covering the dome of the sky. Thus, even for cloudy conditions (reported hourly), it is difficult to distinguish if a single zenith-sky spectrum (sampling frequency is about 1 min) is contaminated by enhanced scattering in the clouds or not. An example is seen in Figure R1.2 (provided in response to other questions from referee #1). In general, to validate a cloud-screening algorithm using independent cloud observations, we need to have cloud observations with similar temporal frequency and be able to determine the cloud distribution (close to zenith direction) and type (e.g., thickness).

B) The choice of the wavelength pair for the calculation of the CI is not be well justified. The authors write 'The 450/550 pair was found to be the most reliable one for the ZSDOAS instruments used in this

work'. It should be made clear in which respect the new choice is better over the other suggested wavelength pairs. (and what is meant with 'reliable'?) In my opinion the new pair is probably even problematic, because the measured radiance at 550 nm strongly depends on the ozone content (at high SZA). Assuming e.g. an ozone VCD of 300 DU and an AMF of 10, this results in an ozone optical depth at 550 nm of about 0.28. This can have a significant effect on the CI and makes the CI dependent on the ozone amount.

We selected this pair based on previous studies and the NDACC UV-visible ozone measurement recommendations. The wavelength regions were chosen to obtain the largest spectral contrast (Rayleigh vs Mie) and also avoid the influence of strong atmospheric absorption features such as ozone. Fig. 2 in Hendrick et al. (2011, https://www.atmos-chem-phys.net/11/5975/2011/) shows an example of an ozone differential slant column fit. Neither 450 nm nor 550 nm show strong absorption features from the fitted species. In fact, 550 nm has been used to calculate the colour index in many published studies as cited in the paper.

In addition, 450 nm and 550 nm are the boundaries of the NDACC-recommended ozone fitting window. So, this colour index pair can easily be used by any other NDACC group member.

We agree with the referee that the ozone content (at large SZA, i.e. SZA > 85°) can affect the colour index. But, the "CI value label" used in this study is limited to small SZA conditions. The changes in CI due to changes in ozone content are small for SZA < 85°. As an example, the colour index with different ozone VCDs is shown in Table R1.1, when cloud optical depth (COD) = 4. The difference caused by a 200 DU ozone VCD variation will only lead to an increase of CI by 0.04 and 0.12 when SZA = 60° and 80°, respectively. On the other hand, if we select any other pair (within the NDACC-recommended ozone fitting window), because of the decreased contrast, the relative changes will be larger than for the 450/550 pair. This information has been added to the manuscript to make this point clearer to readers.

*The 450/550 intensity pair was chosen to obtain the largest spectral contrast (in the NDACC-recommended ozone retrieval window) and also to avoid the influence of strong atmospheric absorption features, such as those of ozone.*

Table R1.1. Colour index (450 nm/550 nm) values simulated (using SCIATRAN) with different total column ozone and SZA values.

| Ozone VCD [DU] | SZA | | |
| --- | --- | --- | --- |
| | 60° | 80° | 90° |
| 300 | 0.97 | 1.13 | 1.57 |
| 500 | 1.01 | 1.25 | 2.18 |

C) The effect of the surface albedo is not discussed. I would expect that it has a systematic influence on the CI. At high latitude sites the surface albedo changes strongly over the year. The authors should discuss this effect and should explain how they deal with the variability of the surface albedo.

Yes, surface albedo has a systematic effect on the CI. But for Eureka, most measurements with SZA < 80° are made in the summer (see Figure R1.1 below). In addition, the CI calibration method used in this study finds the bottom of the "cloudy envelope" by using the smallest simulated CIs. The simulations have different surface albedos to represent summer (0.06, typical of soil and water) and spring/autumn (0.95, typical of snow) conditions. Increased surface albedo will increase the CI at a given SZA. To further quantify the impact of changing albedo, the simulated CIs are shown in Table R1.2 below. For example, when SZA = 80°, changing the albedo from 0.06 to 0.95 will only change the CI by 0.04. In general, this surface albedo effect has a smaller impact on CI values than changes in the ozone column. However, neither the albedo nor the column ozone dependence will make a fundamental difference to the proposed CI calibration method. More details regarding the effects of changes in TCO and albedo on the CI have been included in a new Appendix A.

[Figure]

Figure R1.1. Histogram of zenith-sky measurements at Eureka made with SZA < 80° over the period 2010 to 2017.

Table R1.2. Colour index (450 nm/550 nm) values simulated with different surface albedo and SZA values.

| Surface albedo | SZA | | |
|---|---|---|---|
| | 60° | 80° | 90° |
| 0.06 | 0.97 | 1.13 | 1.57 |
| 0.95 | 1.02 | 1.17 | 1.69 |

D) The SZA dependence (e.g. in Fig. 2) of the model simulations and the measurements is very different (especially for the minimum values). The authors should discuss possible reasons for these differences (maybe related to change of albedo during the year)? Also information on the input for the model simulations should be given, especially the ozone VDD and the surface albedo used for the simulations. In Fig. 1 several jumps are seen for the simulation results. What is the reason for these jumps? They seem to be not realistic.

Following Wagner et al. (2016), the minimum CI values are not from a single simulation. They are defined by the lowest CIs that have been simulated. We think the SZA dependence referred to by the referee is the structure of the density plot (i.e. the cloudy branch; the dots with density > 0.5). However,

this structure is a general pattern of CIs in cloudy conditions; the median values of the cloudy CIs. Thus we are not expecting them (median and minimum values) to have similar SZA dependence. The details were described in Section 3.1. Information on the inputs for the model simulations are included in the Fig. 1 caption.

*Figure 1. Colour index as a function of solar zenith angle. The measurements are from the UT-GBS in 2011, colour-coded by the normalized density of the points. Colour lines are examples of radiative transfer model CI simulations, using a surface albedo of 0.06 and the MPIC climatology ozone profile (total column ozone = 425 DU). Cloud height and cloud optical density (COD) indicated in the legend.*

The jumps in Fig. 1 are due to increased errors in the RTM when simulating large COD and large SZA. We noticed this issue, but it does not affect our calibration algorithm. This is because the simulations for large CODs were used to find the lower limit (lower boundary) of the "cloudy envelope". Thus, the artificially increased CI values (for very large CODs) are not used in the algorithm (we also removed some lines for very large CODs, i.e. COD > 12).

E) In Fig. 1 it is seen that high clouds can have very similar CI as clear sky observations. The authors should check if this result is reasonable. If this simulation results are correct, I have some doubts about the ability of the algorithm to detect high clouds. These clouds might have a considerable effect on the ozone measurements.

High clouds can have similar CI as clear sky obervations at large SZAs (i.e. > 85°). At large SZAs, the cloud index has a complicated dependence on cloud height. The CI has even been used to detect polar stratospheric clouds (Sarkissian et al., 1991). However, the strong height dependence is only relevant for SZA > 85°. For example, comparing the two purple lines in Figure 1, one for 1-3km, COD=1 and the other for 6-8km, COD=1, they are on top of each other. So for small SZA, clouds at 1-3km and 6-8km have almost the same CI. These two lines are different from the clear sky CI (the green line). For large SZA, as discussed in response to previous questions, because of the TCO, albedo, and cloud height effects, the CI value is not used for cloud identification. This information was provided in Section 3.1 and 3.4. Additional details about RTM simulations are included in Appendix A.

F) The authors skip individual measurements, which are indicated as cloudy. I am not sure if this is a good procedure, because it leads to a variable selection of measurements (different numbers, different SZAs), which can have a systematic effect on the derived average ozone results. Also, if only a small number of measurements remains, the total uncertainty might increase. The authors should investigate how the selection of measurements affects the derived average O3 VCD. What is the minimum number of required measurements in a sequence? There is another, related point: it is written, that in some cases the SZA range of the selected measurements is shifted from the standard range (86-91°). How large is the maximum shift of the SZA range? For which situations is a shift applied? How does the shift affect the ozone results?

The selection of measurements in a sequence and the quality control applied to the Langley plots were previously described in the manuscript (Sections 3.3 and 3.4):

*In this work, for each twilight, ozone dSCDs in the NDACC-recommended SZA range (86° to 91°) were selected, when those dSCDs were available. Otherwise, to adapt to the high-latitude condition, the nearest available 5° SZA range was used (Adams, 2012). For quality control purposes, any fit with less than eight measurements or with a coefficient of determination ($R^2$) less than 0.9 was discarded.*

*When cloud-affected spectra have been removed, the same criteria are applied to the cloud-screened Langley plot as apply for the conventional Langley plot (e.g., requires nine data points and $R^2$ >0.9).*

In general, the same quality control criteria are applied to both the conventional Langley plot and the cloud-screened Langley plot. The minimum number of measurements and $R^2$ requirements (to make a Langley plot) are based on Fraser (2008) and Adams (2012). These common quality control criteria, which are shared between the conventional Langley plot and the cloud-screened Langley plot, can ensure good quality for the derived ozone. In fact, those cloud contaminated data (ozone dSCDs) in the Langley plot will introduce more uncertainty. Figure R1.2 shows an example of the Langley fit results with and without using cloud-contaminated data; by removing the cloud contaminated data, the $R^2$, the estimated RCD (the intercept of the fit), and the estimated errors of the slope are all improved. The Langley fits for the morning have an RCD of -2.4e19 (for both the conventional Langley plot and the cloud-screened Langley plot), while the cloud-screened evening RCD (-2.1e19) is 5% lower than the unscreened RCD. Theoretically, the morning and evening RCDs should be the same (from the same reference spectrum). The unscreened morning and evening ozone from UT-GBS are 288 DU and 300 DU, respectively. The corresponding cloud-screened values are 290 DU and 299 DU. The same day Brewer

morning and evening averaged measurements were 298 DU and 299 DU, respectively. The weather record for this day provided by the Eureka Weather Station was "Mainly Clear" from 2:00 to 23:00.

[Figure]

Figure R1.2. Langley plots for measurements made on August 17, 2010 (p.m.). Red symbols show the data without using the cloud-screen algorithm. Blue symbols show the data with the cloud-screen algorithm applied.

The shifting of the SZA fitting window is necessary to produce summer measurements at Eureka. The Langley plots in Figure R1.2 were made using the nearest available 5° SZA range (81-86°). Figure R1.3 below shows the year-round SZA in Eureka. If the NDACC-recommended SZA window (86-91°) is strictly applied in the Langley plot, ozone measurements would only be available for about two months per year at Eureka. Fitting with a lower SZA range will increase the errors due to slant column fitting, but reduce the errors from the AMF calculation (Hendrick et al., 2011). We expect this SZA shift may create some systematic changes in the retrieved ozone, however, the UV-visible TCO dataset has been evaluated through comparisons with satellite and ground-based measurements (e.g., Fraser et al., 2008, Adams et

al., 2012). For example, they show good agreement with Eureka Brewer TCO (mean bias relative to Brewer within 1%).

[Figure]

Figure R1.3. Maximum (red) and minimum (blue) SZAs in Eureka throughout the year, calculated from 2007. The grey shaded region indicates polar night, when the sun does not rise above the horizon, while the yellow shaded area indicates the time of year when the sun is continuously above the horizon. The horizontal dotted lines indicate 86° and 91°, the NDACC-recommended range of SZAs in the calculation of VCDs. Figure from Fraser (2008).

G) The effect of instrument degradation should be addressed. The authors write that in particular the differences in the calibration for the GBS instrument might be related to instrumental changes. The occurrence and strength of changes in the instrumental properties should be stated. Also gradual long term degradation should be investigated.

We think the referee is asking two questions: the first one is about the instrument changes, and the second one is about the long-term instrument degradation. The major changes to the instrument (including FOV changes and integration of a solar-tracking system) are described in Section 2.1. The details of instrument changes and history are available and summarized in Zhao (2017). The following information has been added to the manuscript (Section 2.1 and Section 3.1):

*The shifting of the calibration factor in 2013 is due to the fact that a 10 m slit-to-spot fibre bundle replaced the old 1 m single fibre. The shift in 2017 is due to a 200-grit UV diffuser that was used to attenuate the light signal (to enable MAX-DOAS measurements). Details about all instrument upgrades are provided in Zhao (2017).*

The performance of the instruments is evaluated every year by performing laboratory calibration and tests, including dark current, stray light, polarization, and instrument effects measured yearly and corrected. These tests and the results are documented in Farahani (2006), Fraser (2008), Adams (2012), and Zhao (Zhao, 2017). The GBS ozone data have been used in multiple satellite validation studies (Fraser et al., 2008; Adams et al., 2012; Bognar et al., 2018), and no obvious data degradation has been found in those comparisons.

Minor points:

1) Can the authors explain, for which atmospheric conditions measurements fall into the category 'intermediate'?

The "intermediate" category follows the idea proposed by Gielen et al. (2014). The CI label is used to detect changes in the visibility, and the "intermediate" category represents sky conditions with slightly decreased visibility. For the Arctic, where aerosol pollution is rare, the "intermediate" category applies to sky conditions with thin clouds or moderate aerosol. This information has been added in Section 3.1:

*Following Gielen et al. (2014), we also categorize the calibrated CI values into three regimes as shown in Figure 2b: (1) cloudy, when $CI_{cal}(SZA) < CI_{COD=1.5}$, (2) clear, when $CI_{cal}(SZA) > CI_{visibility=50km}$, and (3) intermediate, when $CI_{COD=1.5}(SZA) < CI_{cal}(SZA) < CI_{visibility=50km}(SZA)$, which represents sky conditions with slightly decreased visibility, typically due to thin clouds or moderate aerosol.*

2) In several parts of the paper, the cloud effects are referred to as 'random', e.g. in the abstract. In other parts, e.g. on page 3, line 17 it is stated that 'This leads to a random uncertainty of 3.3% for TCO calculated using the NDACC ozone AMF LUT between 86-91_ SZA.' Then in the next sentence it is written 'In fact, clouds are the largest source of random uncertainty in ZS TCO.' In my opinion, cloud effects are systematic. Of course, depending on the cloud type, they might

have different effects on the derived O3 VCD. Thus they can indeed introduce a random

component. The authors should discuss these aspects in more detail.

This work uses the NDACC ozone AMF LUT to retrieve ozone total column. Thus, to avoid confusion, we

followed Hendrick et al. (2011) who categorize the uncertainty due to clouds as random uncertainty.

The NDACC UV-visible total column ozone error budget is provided in Table 4 of Hendrick et al. (2011).

Clouds are not accounted for in the NDACC ozone AMF LUT calculations. Due to their varying cloud

properties (height, COD, ice or water content), their impact on the retrieved total column has random

behaviour. For example, although COD = 3 and COD = 6 clouds can both lead to a bias in the retrieved

TCO, the magnitude of the shift is different. In general, unless the clouds can be differentiated into a few

categories (e.g., based on their impact on TCO), it is difficult to quantify the systematic uncertainties due

to clouds. We think the uncertainty categories used in Hendrick et al. (2011) are reasonable in the

context of this work.

3) In Fig. 3 the fitted curve seems to be not a pure Gaussian. Please provide details of the applied fit

function.

The fitting was done using a MATLAB Gaussian model:

https://www.mathworks.com/help/curvefit/gaussian.html

The model provided by MATLAB can be used for multi-peak fitting, but here we only used one peak. For

example for SAOZ 2013 data, the fitting function and results are:

General model Gauss1:

$f\_gaus(x) = a1*exp(-((x-b1)/c1)^2)$

a1 =     37.35

b1 =     0.846

c1 =    0.1475

Some typos in the text and Figure 3 caption have been corrected.

4) Fig. 3: which SZA are included in these results?

These results are from measurements made with SZA < 85°. This information was previously provided in the caption of Fig. 3:

*Note that any measurements with solar zenith angle (SZA) > 85° have been removed in this calibration process.*

5) Title: maybe add 'ground based' between 'on' and 'UV'?

Done.

6) Introduction: on page 3, lines 9-10, also the following reference might be included: Erle F., Pfeilsticker K., Platt U, On the influence of tropospheric clouds on zenith scattered light measurements of stratospheric species, Geophys. Res. Lett., 22 , 2725- 2728, 1995.

This reference has been added.

7) On page 5 it is written: 'Due to the decreased resolution at the edge of CCD, the ozone differential slant column densities (dSCDs) were retrieved in the 450-545 nm window, instead of the NDACC recommended 450-550 nm window.' The Chappuis ozone absorption has no fine spectral structures. Is a high spectral resolution really needed for the ozone analysis in the visible? Maybe the NDACC window can still be used?

Because of the design of the spectrometer (there is a 10° angle between the detector normal axis and the main optical axis of the focusing mirror), the resolution of spectra is not uniform across the CCD (more details can be found in Figure 2.4 of Zhao, 2017). The resolution decreases from 0.9 nm across the centre of the CCD to 1.2 nm at the edge. Although there is no fine structure from 545 to 550 nm, the DOAS fitting will be affected by this poor resolution on the CCD edge, which will reduce the quality of the dSCDs.

8) Fig. 2: Why have measurements for SZA > 85° been removed?

This figure is intended to illustrate the CI calibration. The CI absolute value calibration is done by using data with SZA < 85°. For any measurements made with SZA > 85°, the cloudy CI and clear-sky CI are

difficult to separate. Since measurements with SZA > 85° were not used in the calibration, they were removed from the figure. This information is provided on page 8, lines 27-30 (AMTD version). The caption has been modified to make this point clearer to the reader.

*Note that any measurements with solar zenith angle (SZA) > 85° have been removed in this calibration process, and are not shown here.*

9) Section 3.2: How do the results based on the temporal variation agree with the results derived from the CI threshold method?

These two methods were used to identify the sky conditions. In general, the CI remains stable for clear sky, aerosol, and full cloud cover conditions. However, in the presence of broken clouds, the CI can decrease when a cloud passes over, due to enhanced Mie scattering (Wagner et al., 2016). The temporal variation method is used to detect scattered (broken) clouds (for all available SZAs), while the CI threshold method is used to detect clear sky and full cloud cover (only for SZA<85°). These two methods could not be compared directly.

10) Section 3.3: It is written that 'The inclusion of ozonesonde data in the AMF calculations improves the results, especially under vortex conditions (Bassford et al., 2001).' This statement is unclear to me. Is the use of ozone sonde data an addition to the existing NDACC LUT? Is the original NDACC LUT used in this study or and updated LUT?

The NDACC AMF LUT was produced using climatological ozone profiles (based the on TOMS v8.0 dataset), with total ozone columns from 125 to 575 DU (with a step of 50 DU). Thus, the NDACC LUT requires ozone total column input for each day to better interpolate the stored AMFs for the station. This approach is important for the polar regions when measurements were made under vortex conditions. The Eureka ozonesonde data are used to construct the "Day_SZA_O3_col.dat" file, which provides the total column information, as described in Van Roozendael et al. (2009). Some of the details about the use of NDACC LUT are provided below (from Roozendael et al., 2009):

*"An interpolation routine has been developed to extract appropriately parameterized O3 AMFs for the different NDACC stations. Compared to version 1.0, the new version 2.0 of the routine allows AMFs to be*

*interpolated on a yearly basis.The user has also to define the name of the file with day numbers, SZAs and corresponding O3 columns (here called 'Day_SZA_O3_col.dat'; maximum number of lines in this file: 500000) and to give a value to the flag for the interpolation on the O3 column (fixed to 1 if the O3 columns in 'Day_SZA_O3_col.dat' are vertical columns in DU and to 2 if O3 columns are slant columns in molec/cm2 )."*

In short, the use of ozonesonde data is to improve the interpolation of the stored NDACC AMFs for the station in the polar regions. In this work, we used the original NDACC LUT, and followed the original NDACC recommendation.  This information has been added in Section 3.3:

*Following the NDACC recommendation (Van Roozendael et al., 2009), the Eureka ozonesonde profiles are integrated to generate TCO values that are used to create the "Day_SZA_O3_col.dat" file, which is used by the NDACC LUT to interpolate daily AMFs for Eureka.*

11) Page 14, line 18: It is written: 'Theoretically, the cloud-screened TCO datasets (GBSCS and SAOZCS) should have lower random uncertainties than the conventional TCO datasets (GBS and SAOZ).' I am not sure about this statement. One general effect of the cloud filter is that it removes measurements of a sequence. Thus the information content should be smaller than for a complete sequence. Also the selection of measurements becomes variable: e.g. on some days measurements for small SZA, and on other days large SZA might be filtered. This will lead to different biases and probably to an increased 'random' uncertainty.

This question is related to the previous major question (F). Please refer to some of our explanations for that question. In short, this cloud-screening Langley plot method shared the same (strict) quality control criteria as our traditional Langley plot method. The selection of SZA range, minimum number of measurements, and the threshold for correlation coefficient are all the same for the cloud-filtered and the traditional datasets. Removing some cloud-contaminated spectra improves the Langley fitting results. This is also illustrated in Erle et al.  (1995).

The shifting of the SZA fitting window is necessary to produce measurements during summer time at Eureka. If the NDACC-recommended SZA window (86-91°) is strictly applied in the Langley plot, ozone measurements would only be available for about two months per year (See Figure R1.3). The cloud-screening algorithm shared the same dynamic SZA fitting window method as the traditional algorithm.

**References**

Adams, C.: Measurements of atmospheric ozone, $NO_2$, OClO, and BrO at 80˚N using UV-visible spectroscopy, Ph.D Thesis, University of Toronto, Canada., 2012.

Adams, C., Strong, K., Batchelor, R. L., Bernath, P. F., Brohede, S., Boone, C., Degenstein, D., Daffer, W. H., Drummond, J. R., Fogal, P. F., Farahani, E., Fayt, C., Fraser, A., Goutail, F., Hendrick, F., Kolonjari, F., Lindenmaier, R., Manney, G., McElroy, C. T., McLinden, C. A., Mendonca, J., Park, J. H., Pavlovic, B., Pazmino, A., Roth, C., Savastiouk, V., Walker, K. A., Weaver, D. and Zhao, X.: Validation of ACE and OSIRIS ozone and $NO_2$ measurements using ground-based instruments at 80°N, Atmos. Meas. Tech., 5, 927–953, doi:10.5194/amt-5-927-2012, 2012.

Bognar, K., Zhao, X., Strong, K., Boone, C. D., Bourassa, A. E., Degenstein, D. A., Drummond, J. R., Duff, A., Goutail, F., Jeffery, P., Lutsch, E., Manney, G. L., McElroy, C. T., McLinden, C. A., Millán, L., Pazmino, A., Sioris, C. E., Walker, K. A. and Zou, J.: Validation of ACE and OSIRIS ozone and $NO_2$ measurements using ground-based instruments at Eureka, submitted to J. Quant. Spectrosc. Radiat. Transfer, 30 pages, 2018.

Erle, F., Pfeilsticker, K. and Platt, U.: On the influence of tropospheric clouds on zenith-scattered-light measurements of stratospheric species, Geophys. Res. Lett., 22(20), 2725–2728, doi:10.1029/95GL02789, 1995.

Farahani, E.: Stratospheric composition measurements in the Arctic and at mid-latitudes and comparison with chemical fields from atmospheric models, Ph.D Thesis, University of Toronto, Toronto., 2006.

Fraser, A., Goutail, F., Strong, K., Bernath, P. F., Boone, C., Daffer, W. H., Drummond, J. R., Dufour, D. G., Kerzenmacher, T. E., Manney, G. L., McElroy, C. T., Midwinter, C., McLinden, C. A., Nichitiu, F., Nowlan, C. R., Walker, J., Walker, K. A., Wu, H. and Zou, J.: Intercomparison of UV-visible measurements of ozone and $NO_2$ during the Canadian Arctic ACE validation campaigns: 2004-2006, Atmos. Chem. Phys., 8, 1763–1788, doi:10.5194/acp-8-1763-2008, 2008.

Fraser, A. C.: Arctic and midlatitude stratospheric trace gas measurements using ground-based UV-visible spectroscopy, University of Toronto, Canada., 2008.

Gielen, C., Van Roozendael, M., Hendrick, F., Pinardi, G., Vlemmix, T., De Bock, V., De Backer, H., Fayt, C., Hermans, C., Gillotay, D. and Wang, P.: A simple and versatile cloud-screening method for MAX-DOAS retrievals, Atmos. Meas. Tech., 7, 3509–3527, doi:10.5194/amt-7-3509-2014, 2014.

Hendrick, F., Pommereau, J. P., Goutail, F., Evans, R. D., Ionov, D., Pazmino, A., Kyrö, E., Held, G., Eriksen, P., Dorokhov, V., Gil, M. and Van Roozendael, M.: NDACC/SAOZ UV-visible total ozone measurements: improved retrieval and comparison with correlative ground-based and satellite observations, Atmos. Chem. Phys., 11, 5975–5995, doi:10.5194/acp-11-5975-2011, 2011.

Meteorological Service of Canada: MANOBS: manual of surface weather observations, Meteorological Service of Canada, Ottawa, Ont., 2015.

Sarkissian, A., Pommereau, J. P. and Goutail, F.: Identification of polar stratospheric clouds from the ground by visible spectrometry, Geophys. Res. Lett., 18(4), 779–782, doi:10.1029/91GL00769, 1991.

Wagner, T., Beirle, S., Remmers, J., Shaiganfar, R. and Wang, Y.: Absolute calibration of the colour index and $O_4$ absorption derived from Multi AXis (MAX-)DOAS measurements and their application to a standardised cloud classification algorithm, Atmos. Meas. Tech., 9, 4803–4823, doi:10.5194/amt-9-4803-2016, 2016.

Zhao, X.: Studies of Atmospheric Ozone and Related Constituents in the Arctic and at Mid-latitudes, Ph.D Thesis, University of Toronto, Canada., 2017.

---

## Author Comment (AC2) · 10 Jan 2019

**Response to Referee #3:**

Thank you to referee #3 for the helpful comments. Our responses are given below in black with the comments in blue. The new text in the modified manuscript is given in red (italicized).

**Referee #3:**

Major issues: (1) To assign the label of cloudy, clear or intermediate, the variation of O4 along the day is taken into account. I think this criterion can be stricter for GBS instrument than for SAOZ instrument due to their differences in FOV. As can be seen in figure 4 (although please, see technical comments about this figure), it seems that there are more "clear" data in the case of SAOZ than in the case of GBS. I was wondering if this fact could be due to the O4 criterion. In figure 3, it is quite surprising that for year 2011 clear, cloud and intermediate cases are quite close for both instruments but this situation changes considerably for 2013 and it is clearly different for 2017. But both instruments are located in the same observatory, how is possible that the number of clear/cloudy days in 2017 can be that different? Maybe the O4 criterion is too permissive for SAOZ and too strict for GBS? This could also have an effect in the difference on the bias for both instruments when compared to Brewer. If the algorithm is not properly working for SAOZ, some clear days can, in fact, be affected by clouds and that would explain the better agreement between SAOZCS and Brewer than GBSCS and Brewer.

For 2011, the GBS performed measurements from March to August, and SAOZ performed measurements from March to August. So the percentages of clear/cloudy measurements from two instruments were very similar. For 2013, SAOZ performed measurements from March to April; while, GBS performed measurements from March to October. So the difference in the percentage of clear/cloudy measurements in 2013 was due to the different measurement periods. Please note the y-axis on Figure 4 is not number of days, but the percentage of data (spectra) that has been identified as clear or cloudy. For 2017, UT-GBS has measurements from May to September, while SAOZ has measurements from March to October. The 2013 UT-GBS colour index calibration factor change was due to the old 1 metre fibre being replaced by a 10 metre slit-to-spot fibre. The 2017 UT-GBS colour index calibration factor changes are mainly due to the use of an extra diffuser to decrease the signal (to enable MAX-DOAS measurements). These technical details have been added in the paper (Section 3.1). We also agree with the referee that the optimized $O_4$ criteria could be different for these two instruments, but to

make it a consistent comparison, we used the same criteria for both instruments. A more detailed study could be performed in the future to fine tune this criterion.

*The shifting of the calibration factor in 2013 is due to the fact that a 10 m slit-to-spot fibre bundle replaced the old 1 m single fibre. The shift in 2017 is due to a 200-grit UV diffuser that was used to attenuate the light signal (to enable MAX-DOAS measurements). Details about all instrument upgrades are provided in Zhao (2017).*

(2) To be sure that the effect you observe in the bias when CS is applied to GBS TCO is only due to the presence of clouds, have you take into account that most of cloudy days happen out of the summer?

We have taken this potential seasonal effect into account. We divided the data into summer and spring/fall by using the largest available SZAs, and compared the clear-cloudy differences from these two periods. The summer period is defined as having the largest SZA of the day less than 85° (May to August). In general, when only summer data are included, the impact of the cloud-screening algorithm can be clearly seen. Figures R3.1 and R3.2 are similar to Figure 5, but present data divided into spring/autumn and summer using the largest SZA in the Langley plot.

In general, from these tests, we confirmed that:

1)  The clear-cloudy difference in summer is statistically significant, regardless of whether Brewer or MERRA-2 is used as a reference.

2)  If we use MERRA-2 as a reference, the clear-cloudy difference in spring and autumn data is clear. But if we use Brewer as a reference, the clear-cloudy difference in spring and autumn is not significant (due to limited coincident measurements). For example, for Brewer vs. GBS in spring and autumn, we only have 33 coincident measurements in cloudy conditions.

3)  The proposed cloud-screening algorithm uses three sky-condition labels (CI value label, CI smoothness label, and $O_4$ smoothness label). For spring-time (when SZA >85°), the CI value label is not available. Thus, the efficiency of the cloud-screening algorithm is higher in summer than in spring and autumn.

Some of this information has been added to the paper (Section 4.1.2):

*Since cloudy days mostly appear in the summertime, sensitivity tests were performed with the dataset divided into summer and spring/autumn periods to assess whether there was any seasonal bias. In general, we found that the clear-cloudy difference is still statistically significant in summer, no matter which reference is selected (Brewer or MERRA-2). For spring/autumn, the clear-cloudy difference is statistically significant only when MERRA-2 is used as the reference, but not if Brewer is used as the reference due to the limited number of Brewer measurements given the large SZAs in spring and autumn).*

[Figure]

Figure R3.1. Same as Figure 5, but only including spring and autumn data (when daily maximum SZA > 85°).

[Figure]

Figure R3.2. Same as Figure 5, but only including summer data (when daily maximum SZA < 85°).

What SZA do you use to calculate TCO at summer?

For summertime, when the NDACC-recommended SZA range was not available, we used the nearest available 5° SZA range. This information was previously provided in the manuscript. For example, on May 1, the SZA is in the range of 65° to 85°. Thus, we will use measurements made from 80° to 85° in the Langley plot.

Could the observed bias to Brewer have some to do with the major weight of summer days when you eliminate the cloudy days?

We agree with referee that the observed bias to Brewer may be to its greater weighting towards summer days. However, the bias due to Brewer measurements is inevitable for several reasons. First, the Brewer had limited springtime measurements (it only provides measurements when SZA < 82°, as stated in the manuscript). Second, the Brewer cannot perform measurements when heavy clouds block the solar beam. Thus, Brewer measurements are biased to summer and clear-sky conditions. This is the reason we included MERRA-2 in this work. For any study that only uses Brewer data to compare with NDACC-type UV-vis measurements, it is hard to assess the cloud impacts.

Minor issues: (1) Due to the high latitude of the observatory it is not possible to have DOAS measurements along the entire year. Please, in the description of the instrument include what is the annual period of measurements. From figure 4 and from data along the text it seems that the period is late winter to late autumn? It would be nice to know the months when DOAS and Brewer can measure.

The Brewer typically can provide measurements from April to August, while GBS and SAOZ can provide measurements from March to September. This information has been added to in Section 4.

*The Brewer instrument at Eureka typically makes measurements from April to August, while UT-GBS and SAOZ can provide measurements from March to September.*

(2) Section 4.1. Why the current agreement to Brewer and GBS is better in this work than in the previous work by Adams et al.?

The result (-1.4%) in Adams et al. (2012) was based on measurements from 2004 to 2011. For the current study, the result (-0.23%) is based on measurements from 2010 to 2017. There are several possible reasons for the improvement, such as year-round variability, improvement due to new NDACC ozone LUT, and more summertime measurements in the current datasets. During the 2004 to 2006 period, only springtime measurements were available. For the 2007 to 2009, the instrument was using a different grating for the summer measurements. In general, we could not apply the new cloud-screening algorithm to the data before 2010, thus we did not include 2004 to 2009 data in the current work. The 2004-2017 GBS data were reprocessed and used in a satellite validation paper (Bognar et al., 2018,

submitted to JQSRT). In that work, we find that for the 2004-2017 period, the mean relative bias between GBS and Brewer is -0.9%, which is closer to the number reported by Adams et al. (2012). Also, Adams et al. (2012) defined the mean relative differences ($\Delta_{rel}$) as:

$$\Delta_{rel} = 100 \times \frac{1}{N} \sum_{i=1}^{N} \frac{(M_{1i} - M_{2i})}{(M_{1i} + M_{2i})/2},$$

where N is the number of measurements, $M_1$ and $M_2$ are sets of coincident measurements. In Figure 5 (AMTD version), the mean relative difference was defined as:

$$\Delta_{rel} = 100 \times \frac{1}{N} \sum_{i=1}^{N} \frac{(M_{1i} - M_{2i})}{M_{2i}},$$

where $M_1$ was UT-GBS (SAOZ), and $M_2$ was Brewer (MERRA-2), indicated by the y-axis label (see the AMTD version).

To make this study directly comparable with Adams et al. (2012), we have revised Figure 5 and the relevant numbers (using the same $\Delta_{rel}$ definition as Adams et al. (2012)). These changes do not affect the conclusions.

*Following Adams et al. (2012), the agreement between sets of coincident measurements ($M_1$ and $M_2$) was evaluated using the mean relative difference, defined as*

$$\Delta_{rel} = 100 \times \frac{1}{N} \sum_{i=1}^{N} \frac{(M_{1i} - M_{2i})}{(M_{1i} + M_{2i})/2}, \quad\quad (4)$$

*where N is the number of measurements.*

(3) Taking into account the current results, it seems that in the case of Hendrick et al., not all the observed discrepancies between DOAS and Brewer were due to the temperature dependence of XS used in Brewer analysis or in this work the Brewer analysis takes into account this dependence?

The Brewer data used in this work were processed by the standard Brewer algorithm. The temperature dependence due to the ozone cross section does exist in this Brewer dataset. This temperature dependence is different from instrument to instrument. Currently, we do not have an estimated temperature dependence factor for the Brewer instrument used in this study, so no temperature correction was applied.

The temperature dependence of Brewer data also depends on the location of the site. For example, if we assume the temperature dependence of a Brewer is 0.1%/K (as reported in previous studies, e.g.

Kerr, 2002), for a year-round 15 K stratospheric effective ozone temperature variation, the temperature dependence introduced by seasonal changes in TCO will be 1.5%. However, for Eureka, the Brewer only performs measurements from April to August, and so the temperature effect at Eureka is expected to be smaller (compared to year-round mid-latitude measurements). We calculated the effective ozone temperature (based on the method shown in Zhao et al., 2016) for 55°N and 75°N using ozone and temperature profiles from the Max Planck Institute for Chemistry (MPIC, Brühl and Crutzen, 1993) climatology to illustrate this. As shown in Figure R3.3, the estimated temperature-induced bias in Brewer TCO at 75°N is only 0.9% (while for 55°N, this is increased to 1.4%). Thus, to further separate the temperature dependence, cloud effect, and other potential seasonal effects, we will need more accurate temperature and pressure profile measurements or modelled values for Eureka.

[Figure]

Figure R3.3. Simulations of year-round effective ozone temperatures ($T_{eff}$) at two latitudes based on climatological ozone and temperature profiles.

Do you observe also the same seasonal difference (taking into account that you cannot observe the entire spring and fall at 80°N) that Hendrick et al. in the bias against the Brewer?

The seasonal difference between UV-vis TCO and Brewer TCO at Eureka is weaker than reported values measured at mid-latitude sites (e.g., Hendrick et al., 2011). Figure R3.4 shows the ratio of SAOZ and Brewer TCO over the period 2010 to 2017.

[Figure]

Figure R3.4. SAOZ/Brewer total column ozone (TCO) ratio as a function of day of the year for the period 2010 to 2017.

(4) Section 4.1.1, please indicate at any part of the text that the weather classification used here and in figure 5 is made by using meteorological data. If not, it is a little confusing.

The following text has been added in Section 4.1.1:

*The weather classification used here and in Figure 5 is based on hourly observations of sky conditions made by a meteorological technician at Eureka.*

Technical issues:
(1) Figure 4. Please, unify ticks in the horizontal axis. The lower graph is different from the previous ones and this makes very difficult to see properly the measurement periods. Grid in the middle of each year would be also very helpful. Colours in the legend are not coincident with the ones in the graphs. As GBSCS or SAOZCS are over imposed to GBS and SAOZ respectively, it seems that there are more data

for the CS filtered data than without any filter. This is a little bit confusing at first, I am not sure that it can be addressed, maybe using hollow symbols for CS case? If possible it would be nice a greater graph.

Figure 4 has been revised as suggested.

(2) Sometimes the DOAS instrument GBS is called UT-GBS, please unify nomenclature along the text.

UT-GBS has been adopted throughout.

**References**

Adams, C., Strong, K., Batchelor, R. L., Bernath, P. F., Brohede, S., Boone, C., Degenstein, D., Daffer, W. H., Drummond, J. R., Fogal, P. F., Farahani, E., Fayt, C., Fraser, A., Goutail, F., Hendrick, F., Kolonjari, F., Lindenmaier, R., Manney, G., McElroy, C. T., McLinden, C. A., Mendonca, J., Park, J. H., Pavlovic, B., Pazmino, A., Roth, C., Savastiouk, V., Walker, K. A., Weaver, D. and Zhao, X.: Validation of ACE and OSIRIS ozone and $NO_2$ measurements using ground-based instruments at 80°N, Atmos. Meas. Tech., 5, 927–953, doi:10.5194/amt-5-927-2012, 2012.

Bognar, K., Zhao, X., Strong, K., Boone, C. D., Bourassa, A. E., Degenstein, D. A., Drummond, J. R., Duff, A., Goutail, F., Jeffery, P., Lutsch, E., Manney, G. L., McElroy, C. T., McLinden, C. A., Millán, L., Pazmino, A., Sioris, C. E., Walker, K. A. and Zou, J.: Validation of ACE and OSIRIS ozone and $NO_2$ measurements using ground-based instruments at Eureka, submitted to J. Quant. Spectrosc. Radiat. Transfer, 30 pages, 2018.

Brühl, C. and Crutzen, P. J.: MPIC two-dimensional mode, in The atmospheric effects of stratospheric aircraft, vol. 1292 of NASA Ref. Publ., pp. 103–104., 1993.

Hendrick, F., Pommereau, J. P., Goutail, F., Evans, R. D., Ionov, D., Pazmino, A., Kyrö, E., Held, G., Eriksen, P., Dorokhov, V., Gil, M. and Van Roozendael, M.: NDACC/SAOZ UV-visible total ozone measurements: improved retrieval and comparison with correlative ground-based and satellite observations, Atmos. Chem. Phys., 11, 5975–5995, doi:10.5194/acp-11-5975-2011, 2011.

Kerr, J. B.: New methodology for deriving total ozone and other atmospheric variables from Brewer spectrophotometer direct sun spectra, J. Geophys. Res., 107, doi:10.1029/2001JD001227, 2002.

Zhao, X., Fioletov, V., Cede, A., Davies, J. and Strong, K.: Accuracy, precision, and temperature dependence of Pandora total ozone measurements estimated from a comparison with the Brewer triad in Toronto, Atmos. Meas. Tech., 9, 5747–5761, doi:10.5194/amt-9-5747-2016, 2016.

---

## Author Response (AR2)

**Response to Editor:**

Thank you to the editor for the helpful comments. Our responses are given below in black with the comments in blue. The new text in the modified manuscript is given in red (italicized).

**Editor:**

Comments to the Author:

The paper is generally suitable for publication, however I agree with the referee that there are clear anomalies in the simulated CI values presented in Fig.1. These need to be clarified or fixed before final publication. Please contact the SCIATRAN experts in Bremen to discuss the possible reasons for these anomalies and how to avoid them.

We appreciate the help from the editor and two referees in providing helpful comments to this work, which improved the quality of the results. Following the editor's suggestion, we contacted a SCIATRAN expert in Bremen (Alexei Rozanov). The anomalies were found due to the insufficient discretization of the cloud layer. To solve this, twelve cloud sub-layers are used in the high COD simulations.

Changes made:

1.  A modified Fig. 1 has been made and information about the number of cloud sub-layers are provided in the caption. As explained in response to referee #1, these changes will not affect the algorithm proposed in this study and relevant findings.

[revised manuscript text omitted]